# Microridge-like structures anchor motile cilia

Takayuki Yasunaga[1,6], Johannes Wiegel [1,6], Max D. Bergen [1,6], Martin Helmstädter[1], Daniel Epting[1], Andrea Paolini[1,2], Özgün Çiçek[3,4], Gerald Radziwill [4,5], Christina Engel[1], Thomas Brox[3,4,5], Olaf Ronneberger[3,4], Peter Walentek[1,5], Maximilian H. Ulbrich [1,4] & Gerd Walz [1,4,5✉]

Several tissues contain cells with multiple motile cilia that generate a fluid or particle flow to support development and organ functions; defective motility causes human disease. Developmental cues orient motile cilia, but how cilia are locked into their final position to maintain a directional flow is not understood. Here we find that the actin cytoskeleton is highly dynamic during early development of multiciliated cells (MCCs). While apical actin bundles become increasingly more static, subapical actin filaments are nucleated from the distal tip of ciliary rootlets. Anchorage of these subapical actin filaments requires the presence of microridge-like structures formed during MCC development, and the activity of Nonmuscle Myosin II. Optogenetic manipulation of Ezrin, a core component of the microridge actin-anchoring complex, or inhibition of Myosin Light Chain Kinase interfere with rootlet anchorage and orientation. These observations identify microridge-like structures as an essential component of basal body rootlet anchoring in MCCs.

[1] Department of Medicine IV, University Freiburg Medical Center, Faculty of Medicine, University of Freiburg, Hugstetter Strasse 55, 79106 Freiburg, Germany. [2] Faculty of Biology, University of Freiburg, Schaenzlestrasse 1, 79104 Freiburg, Germany. [3] Pattern Recognition and Image Processing, Department of Computer Science, University of Freiburg, Georges-Köhler-Allee 52, 79110 Freiburg, Germany. [4] BIOSS Centre for Biological Signalling Studies, University of Freiburg, Schänzlestrasse 18, 79104 Freiburg, Germany. [5] CIBSS Centre for Integrative Biological Signalling Studies, University of Freiburg, Schänzlestrasse 18, 79104 Freiburg, Germany. [6] These authors contributed equally: Takayuki Yasunaga, Johannes Wiegel, Max Bergen. ✉email: gerd.walz@uniklinik-freiburg.de

Cilia, attached to most body cells, have been recognized as central signaling hubs that orchestrate development and tissue homeostasis[1–5]. Defective cilia are associated with diverse developmental and degenerative abnormalities, collectively termed ciliopathies that affect millions of patients worldwide[6]. While a non-motile primary cilium decorates most cells, motile cilia are found on specialized epithelia of the airways, the oviducts and the brain ventricles, where they generate a directional fluid or particle flow. Motile ciliopathy disorders cause debilitating human conditions, including hydrocephalus, infertility, laterality defects and airway disease[3].

The genetically tractable *Xenopus* has become an important model to characterize the function of motile cilia and cilia-associated molecules[7,8]. Multiciliated cells (MCCs) of the *Xenopus* epidermis develop from precursor cells that arise from deeper epidermal layers and penetrate the superficial cell layer by radial intercalation[9–11]. Acentriolar deuterosome-dependent biogenesis creates several dozens of basal bodies[12] that are decorated by accessory structures including rootlets before attaching to the apical plasma membrane to form the ciliary axoneme[13]. The motile cilia of the *Xenopus* epidermis engage in metachronal beating to generate a coordinated fluid flow along the longitudinal axis of the developing frog embryo[14].

Metachronicity mandates that cilia of MCCs beat phase-shifted with the same frequency along the direction of the stroke, but synchronously with neighboring cilia in the perpendicular axis[15]. To achieve this spatial and temporal coordination, cilia are progressively polarized in response to hydrodynamic forces and planar cell polarity signaling during multiciliated cell development, while their orientation is locked in mature MCCs[2,16]. Cilia formation and polarization require an intact cortical actin cytoskeleton[17–22], encompassing two layers, a dense apical actin layer, consisting of bundles of actin filaments that surround the basal bodies, and a less dense subapical layer of actin filaments[23]. Disruption of the subapical actin filaments prevents polarization and a metachronal beating pattern.

Ciliary adhesion complexes, containing the Focal Adhesion Kinase (FAK) and other typical focal adhesion molecules, connect the actin filaments of rootlets of neighboring basal bodies and are required for actin-dependent apical transport of basal bodies[24]. Two types of rootlets, primarily composed of the ~225 kDa protein Rootletin, attach to the proximal end of MCC basal bodies, a prominent anterior rootlet located opposite to the basal foot, and a thinner rootlet projecting towards the cytoplasm[13]. The basal foot originates from subdistal appendages, requiring the presence of outer dense fiber 2 (ODF2), KIF3a, and Ankyrin repeat and SAM domain-containing protein 1A (ANKS1A)[25–27]. Rootlet and basal foot play important roles in resisting the mechanical forces imposed on the basal bodies of motile cilia[28,29], but how basal bodies are precisely anchored to the apical plasma membrane remains unknown.

Here we show that nucleation and polarization of cilia coincided with the formation of apical microridge-like structures and development of the subapical actin layer. Inhibition of Nonmuscle Myosin II (NMII) or manipulation of Ezrin prevented formation of these structures and disrupted ciliogenesis, revealing their importance in anchoring basal bodies into their final position.

## Results

**Ciliary rootlets nucleate actin filaments**. The subapical layer of the cortical actin cytoskeleton plays important roles in basal body orientation and generation of the metachronal fluid wave, which requires synchronized and polarized beating of the motile cilia of multiciliated cells (MCCs) of the *Xenopus* epidermis[21–23].

Analyzing the emergence of the cortical actin cytoskeleton, we found that the subapical actin layer followed maturation of the apical actin layer (Fig. 1a). While few subapical actin filaments were detectable at stage 20-21, most basal body rootles were decorated with actin fibers after stage 26, coinciding with the determination of ciliary polarity[2]. In contrast, the apical actin layer was already dense at stage 25 and before nucleation of the subapical actin layer, suggesting that the apical actin web provides important cues for structuring the subapical actin layer. Basal bodies of mature MCCs form a fan-shaped anterior rootlet that is tilted towards the apical plasma membrane, while one or more thin posterior rootlets point towards the nucleus[30]. We noted that at early stages of MCC development (stage 21-22), the anterior rootlet was aligned orthogonally to the apical membrane (Fig. 1b). However, with the appearance of subapical actin filaments at the distal tip, the anterior rootlet started to tilt towards the apical membrane (Fig. 1c–e). In mature MCCs, subapical actin filaments extended from the tip of the anterior rootlet to the apical plasma membrane. The tilting of the anterior rootlet presented itself as an apparent lengthening of the anterior rootlet, when assessed from the apical (x-y) view. Subapical actin filaments, originating from the distal end of the anterior rootlet, appeared to connect anterior rootlets to actin-capturing structures present at the apical plasma membrane. Rootlets devoid of actin filaments were randomly aligned, but became polarized with the emergence of rootlet-associated actin filaments (Fig. 1f). Thus, polarization of the motile cilia of MCCs coincides with the formation of actin filaments nucleated from anterior ciliary rootlets, and tilting with apparent lengthening of the anterior rootlet when assessed from an x-y view (Fig. 1g, h). To confirm the developmental change in rootlet position, serial sections of MCCs were performed at stage 20-21 and at stage 30. We analyzed the rootlet position of single cilia by transmission electron microscopy (Fig. 2A), and generated 3D reconstructions (Fig. 2B) to determine the distance between the tip of rootlets and the plasma membrane as well as the angle formed between basal bodies and rootlets. Rootlet tilting significantly decreased both the angle and the distance between the tip of the rootlet and the plasma membrane (Fig. 2C).

**Fhod3 co-localizes with FAK to ciliary adhesions**. The Focal Adhesion Kinase (FAK) together with other focal adhesion components localizes to anterior rootlets, implying that anterior rootlets form a platform for actin nucleation and focal adhesion-like structures[24]. Formins are important actin-nucleating proteins implicated in ciliogenesis[31]. Since the formin inhibitor SMIFH2 prevented rootlet lengthening, reduced the fluid flow generated by MCCs, and disrupted ciliary polarity (Supplementary Fig. 1), we analyzed the involvement of formins in anchoring basal body rootlets. Using a library of the known 15 formin family members[32], we found that the formin Delphilin co-localized with Centrin at the basal bodies (Fig. 3a), while Formin homology 2 domain containing 1 (Fhod1) surrounded basal bodies in a donut-like fashion, occupying the space in between basal bodies and actin filaments (Fig. 3b). However, only Fhod3 localized to the anterior rootlet (Fig. 3c), and additionally labeled posterior rootlets (Fig. 3d). Truncations of Fhod3 revealed that the N-terminal domains, containing the GTPase-binding and the Formin homology 3 domain, labeled both, anterior and posterior rootlets, while a C-terminal truncation, containing the Diaphanous auto-regulatory and the Formin homology 1 and 2 domains, associated exclusively with anterior rootlets (Supplementary Fig. 2). We next determined the localization of Fhod3 relative to FAK. Fhod3 located in between phalloidin-labeled actin filaments and the FAK-positive domain of the anterior rootlet within the

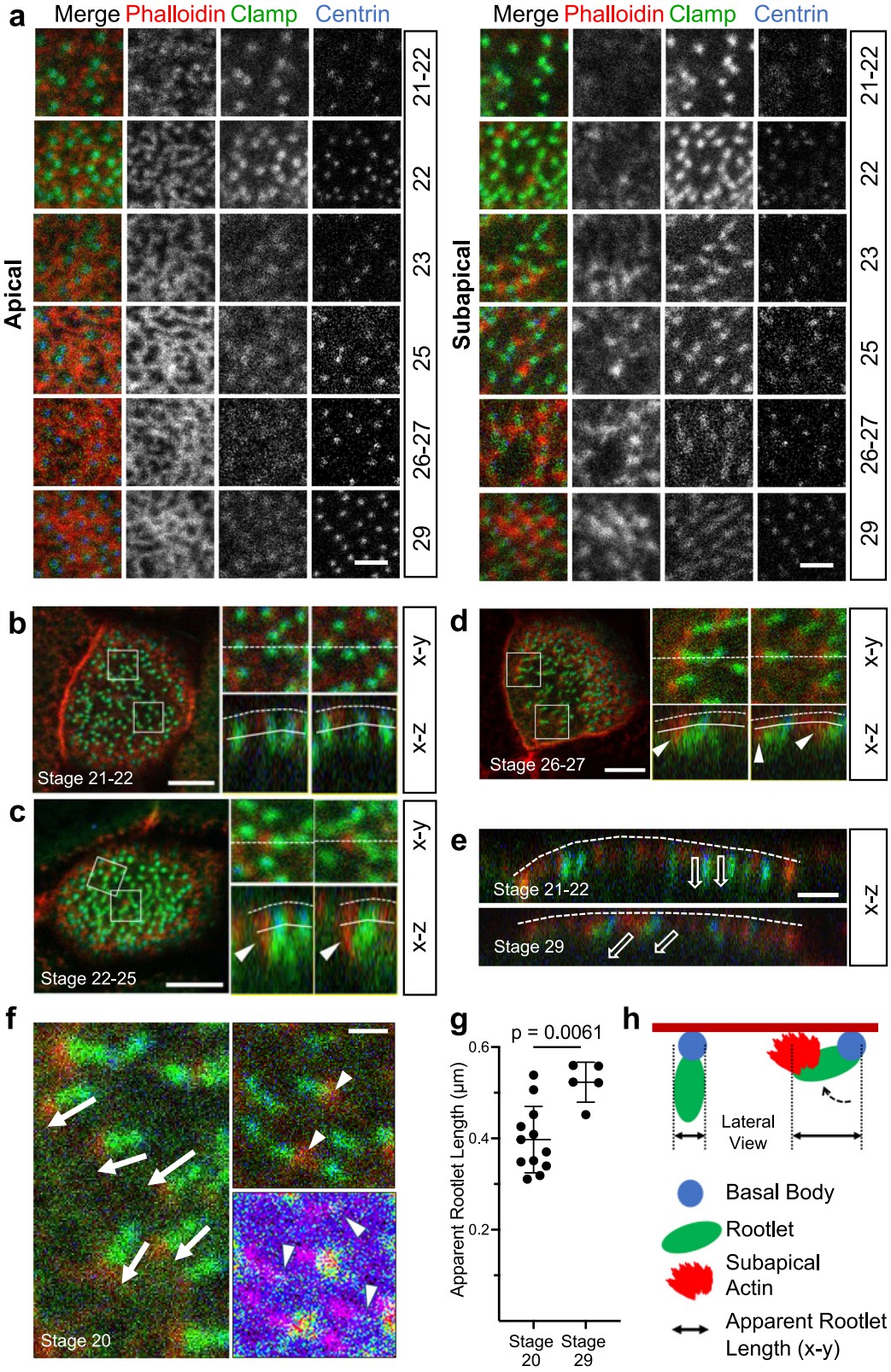

subapical actin layer (Fig. 4, Supplementary Fig. 3), implicating formin family members in nucleating subapical actin filaments at the ciliary rootlet. Depletion of *fhod3*, using three different morpholino oligonucleotides interfered with rootlet tilting, determined by measuring the apparent rootlet length and ciliary polarity (Supplementary Fig. 4). *Xenopus fhod3* depletion primarily affected the subapical actin layer (Supplementary Fig. 5),

further supporting an involvement of Fhod3 in rootlet positioning.

**Multiciliated cells form microridge-like structures**. Since actin filaments arise from the anterior rootlet and project towards the apical cell surface, we searched for an actin anchoring complex at the plasma membrane. We previously described that the loss of

**Fig. 1 Development of the subapical actin in multiciliated cells (MCCs). a** While the apical actin cytoskeleton reaches its final density by stage 25, the subapical actin continues to develop until stage 29 (scale bar, 2 μm). **b** Basal body (ciliary) rootlets, labeled with GFP-Clamp (green), are orthogonally aligned to the plasma membrane at stage 21-22. The right panels represent magnified insets marked by white squares. The dashed line depicts the apical plasma membrane, the solid line separates the apical and subapical actin layer (scale bar, 2 μm). **c** Actin filaments, labeled with phalloidin (red, white arrow heads) start originating from ciliary rootlets (GFP-Clamp) at stage 22–25. The right panels represent magnified insets marked by white squares (scale bar, 2 μm). **d** After nucleation of actin filaments (phalloidin, red), rootlets are tilted towards the apical plasma membrane at stage 26-27. The right panels represent magnified insets marked by white squares. The white arrowheads indicate subapical actin filaments, projecting toward the apical plasma membrane. The dashed white line depicts the apical plasma membrane, the solid line separates the apical and subapical actin layer (scale bar, 5 μm). **e** While rootlets align orthogonally to the plasma membrane (dashed line) at stage 21-22, subapical actin filaments, extending from the tip of the rootlet, connect the rootlets with the plasma membrane at stage 29 (scale bar, 2 μm). **f** Actin nucleation at the tip of the anterior rootlet coincides with rootlet polarization and orientation along the longitudinal body axis (white arrows). Subapical actin filaments (phalloidin, red) are nucleated from the tip of the rootlet (white arrowheads) (right upper insert; Clamp-GFP, green). The pseudo-colored actin heat map depicts phalloidin signals (white arrowheads), originating at the distal end of rootlets (magenta) (right lower insert) (scale bar, 1 μm). **g** Rootlets are laterally elongated at stage 29 as compared to stage 20. Representative data of two independent experiments, depicting the results of 12 cells at stage 20, and 5 cells at stage 29. Each circle represents the average length of rootlets in a multiciliated cell (mean ± SD; Mann–Whitney-U test). Source data are provided as a Source Data file. **h** The diagram explains the apparent elongation of apically imaged (x-y) rootlets after nucleation of subapical actin.

Ezrin disrupts ciliogenesis[33]. Ezrin is an essential organizer of the apical terminal actin web[34], and a component of microridges[35–37]. Microridges are fingerprint-like, F-actin-based structures with poorly defined functions present on epithelial cells[37]. They arise from focal protrusions that confluence to labyrinth-like structures. Transmission electron microscopy (TEM) revealed that MCCs form regularly spaced microridge-like structures (for simplicity now referred to as microridges) (Supplementary Fig. 6). Scanning electron microscopy (SEM) confirmed the maze-like membrane protrusions, surrounding docking basal bodies. While these microridges were largely absent on emerging MCCs at stage 22-23, microridges became increasingly prominent, and were particularly visible at late stages of MCC development (stage 40–42), when cilia started to retract (Fig. 5a)[38]. Analysis of Ezrin in relationship to the apical actin cytoskeleton revealed that Ezrin was diffusely distributed in MCCs during early development (stages 21–23), but started to co-localize with phalloidin-labeled actin bundles at stage 24 (Fig. 5b). A GFP-fusion protein, containing only the FERM domain of Ezrin co-localized primarily with the apical actin filaments, while full-length Ezrin also decorated actin filaments in the subapical layer (Supplementary Fig. 7). Correlation of phalloidin-labeled actin bundles and microridges visualized by SEM revealed that the apical actin bundles detected by light microscopy were identical with microridges depicted by SEM (Fig. 5c, Supplementary Fig. 8). Further analysis demonstrated that the actin filaments arising in the subapical actin layer were connected to the Ezrin-labeled microridges (Fig. 5d), suggesting that microridges anchor the subapical actin-ciliary rootlet complex. Ezrin is known to interact with Fhod1[39]. We found a similar interaction between Ezrin and Fhod3 (Supplementary Fig. 9). Thus, Ezrin appears to serve a dual function, connecting actin filaments with ciliary rootlets as well as anchoring them to the apical plasma membrane.

**Ezrin and Nonmuscle Myosin II (NMII) are required for ciliary rootlet anchoring.** Knockdown of *ezrin*[33] disrupted the structure of the actin cytoskeleton, destroyed the morphology of the microridges, and abrogated ciliogenesis (Fig. 6a and Supplementary Fig. 10a, b). These results support the hypothesis that Ezrin participates in capturing actin filaments that are nucleated from anterior rootlets to anchor cilia into their final position. Confirming the morpholino oligonucleotide results, expression of the C-terminal, actin-binding truncation of Ezrin, likely competing with endogenous Ezrin, shortened rootlet length in a similar fashion, indicative of impaired rootlet tilting (Supplementary Fig. 10c). Mosaic expression demonstrated that the N-terminal FERM domain of Ezrin exerted dominant-negative effects in a cell-autonomous manner (Supplementary Fig. 11).

Since NMII-mediated apical constrictions drive microridge formation[40], we examined the impact of ML7, a selective MLCK inhibitor that prevents Calmodulin-dependent MLCK and NMII activation. ML7 affected basal body distribution and rootlet orientation (Supplementary Fig. 12). Analysis of the MCC surface by scanning electron microscopy revealed that ML7 alters the microridge architecture (Fig. 6b). To quantify the changes, we used U-Net, a deep-learning algorithm for single-cell segmentation[41]. ML7, while increasing the cell surface area, decreased the microridge branching points and thickness, reducing the area covered by microridges (Fig. 6c). To demonstrate that microridges help to maintain basal body orientation, we employed optogenetics to disrupt the anchorage between basal body rootlets and microridges after stage 20. Since Ezrin truncations exerted a dominant-negative effect (Supplementary Figs. 10 and 11), we created an Ezrin fusion protein that dissociates upon exposure to far-red light (740 nm) and re-associates at red light (660 nm), using the phycocyanobilin-dependent PHYB/PIF system[42,43]. To test this approach in MCCs of the *Xenopus* epidermis, we utilized the GFP-PIF (cytoplasmic) and PHYB-mCherry-CAAX (membrane-associated) fusion proteins (Fig. 7a)[42]. Upon exposure to red light, the GFP-PIF fusion protein correctly localized to the plasma membrane, while exposure to far-red light reversed the interaction of both proteins. Fusing the N-terminal FERM domain of Ezrin to PIF, and the C-terminal domain (CTD) to PHYB, we used this system to keep Ezrin assembled until stage 20 by exposure to red light (Fig. 7b). Subsequently, we either dissociated Ezrin (exposure to far-red light), or maintained the association by exposing MCCs to red light. This approach revealed that an interference with Ezrin function even at later developmental stages affected basal body rootlet tilting and polarity (Fig. 7c). Our findings suggest that microridge-like structures are formed during early MCC development, which facilitate the assembly of Ezrin-containing complexes and basal body rootlet anchoring (Fig. 8).

**Discussion**
Microridges form on the surface of MCCs. Microridges, also termed microplicae, are complex three-dimensional structures that arise from aggregation of actin-based, finger-like membrane protrusions to form a labyrinth-like pattern[44]. Although microridges are found on the surface of many tissues[44], including the *Xenopus* epidermis[45,46], mammalian kidney epithelial cells[47,48] and Fallopian tube cells[49], they have been most extensively characterized in fish epithelial systems[35,36,50–52]. While the membrane protrusions identified on the surface of multiciliated cells of the *Xenopus* epidermis share many features with the

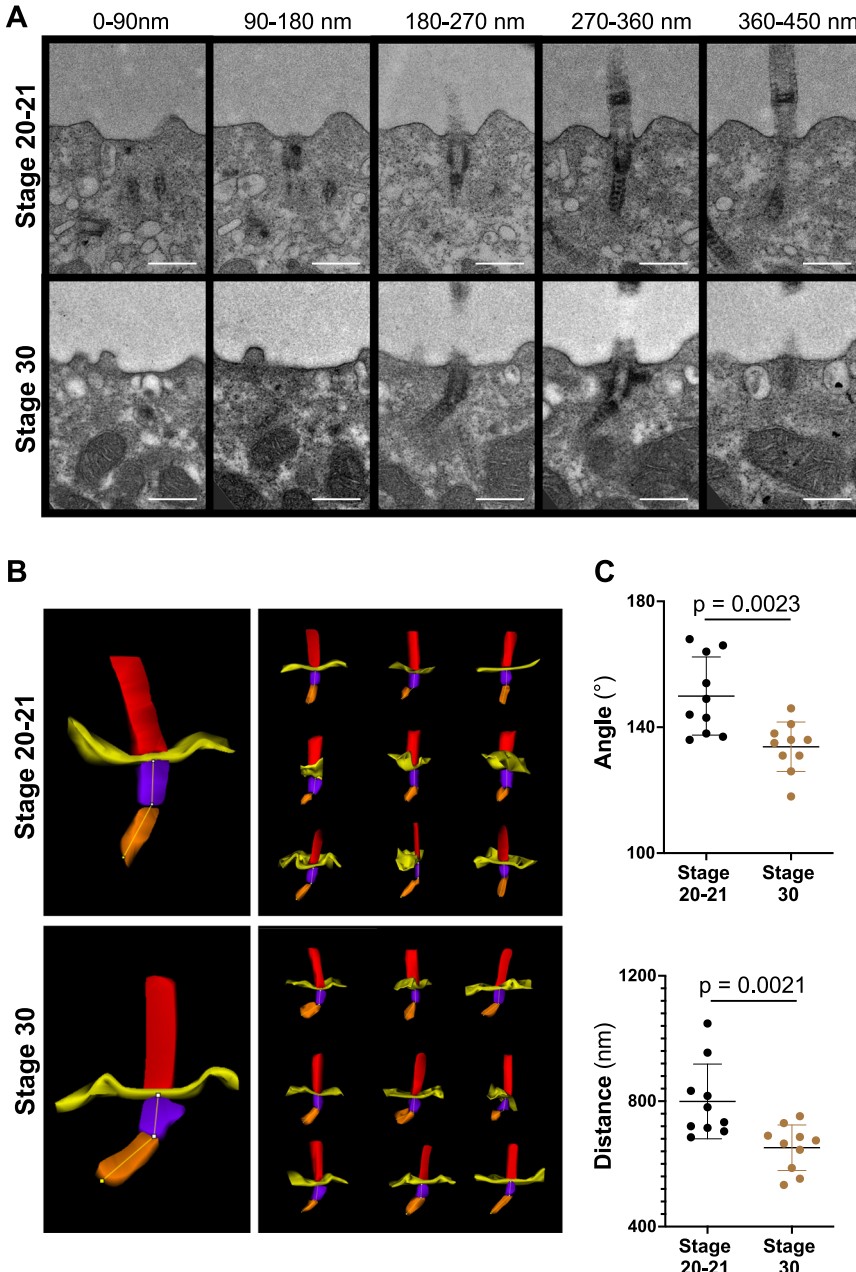

**Fig. 2 Basal body rootlet tilting between stage 20-21 and stage 30. A** Serial sections of multiciliated cells were performed from 7 different cells (one embryo) fixed at stage 20-21 and from 5 different cells (one embryo) fixed at stage 30. Ten cilia were analyzed at each stage by transmission electron microscopy (scale bars, 500 nm). **B** 3D reconstructions were performed, using Reconstruct (Movie 1). **C** The minimal angle between rootlets and basal bodies (upper panel), and the minimal distance from the tip of the basal body rootlet to the plasma membrane (lower panel) significantly decreased between stage 20-21 and stage 30 (see Movie 2 for the applied workflow) (mean ± SD; Mann–Whitney-*U* test). Source data are provided as a Source Data file.

microridges of the zebrafish skin ionocytes, we used the term "microridge-like structures" to acknowledge the unique cortical actin cytoskeleton that differentiates MCCs from most other epithelial cells. The functions of microridges are not well defined; however, their widespread distribution suggests an involvement in multiple tissue functions. Recent observations revealed that NMII-mediated actomyosin constrictions drive microridge formation[40,52]. Motor activity of NMII is triggered by MLCK-mediated phosphorylation of myosin light chain; both RhoA/ROCK and $Ca^{2+}$ bound to Calmodulin (CaM) can activate MLCK and initiate actomyosin contractions[53]. Inhibition of MLCK by ML7, a cell-permeable Calmodulin antagonist, resulted

in abnormal microridge formation and basal body spacing. Although this observation does not establish a causal relationship, the formation of the apical actin network and microridges prior to subapical actin nucleation suggest that intact microridges are required for subsequent basal body polarization.

**Microridges are actin-anchoring membrane protrusions.** Several components have been identified that localize to the microridge compartment, including membrane-tethered mucins, Cortactin, Paxillin, NMII and Ezrin[34,35,44,54]. Ezrin binds $PIP_2$ through its FERM domain and actin through its C-terminal

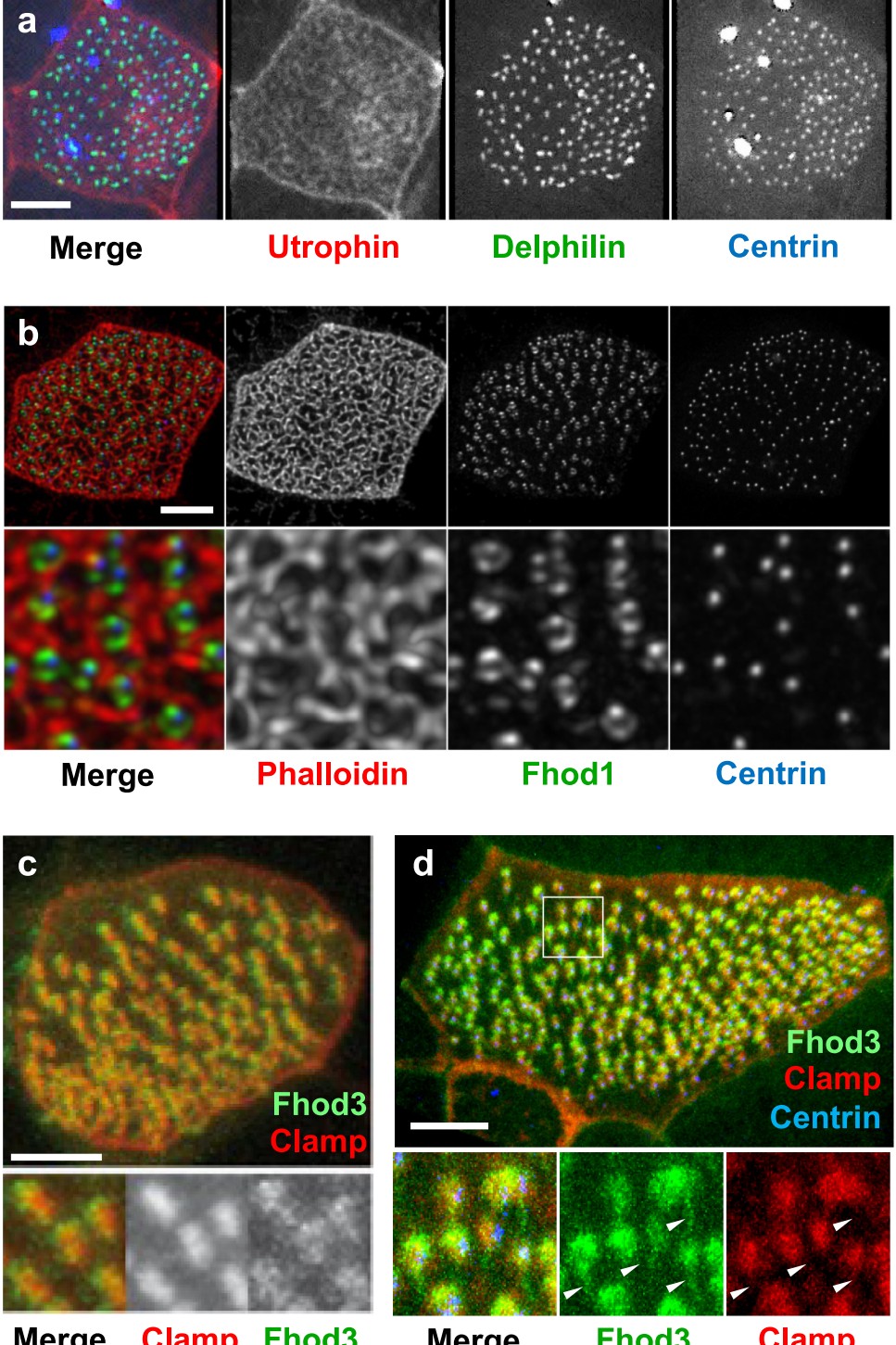

**Fig. 3 Localization of Fhod3 in multiciliated cells (MCCs). a** Delphilin, labeled with GFP, co-localizes with BFP-labeled Centrin to the basal bodies of MCCs. Actin filaments were visualized by co-expression of RFP-Utrophin. **b** Fhod1 (GFP-Fhod1) assumes a position in between surrounding actin filaments (phalloidin, red) and basal bodies (BFP-Centrin). **c** GFP-Fhod3 localizes to the tip of the RFP-Clamp-labeled anterior rootlet. The lower panel shows magnified Clamp- and Fhod3-labeled images. **d** Fhod3 associates with anterior and posterior rootlets. The posterior rootlets (arrowheads) are oriented orthogonally to the plasma membrane. Depicted are diagonal views after 3D reconstruction of confocal images (IMARIS). Images were obtained at stage 29-30; scale bars, 5 μm).

domain[55]. Accumulation of $PIP_2$ in the inner lipid layer of outward concave membranes[56] can facilitate the recruitment and accumulation of Ezrin within microridges; alternatively, Ezrin has been shown to bind to cytoplasmic domains of several membrane proteins[57]. We found that the FERM domain of Ezrin localized to the actin bundles outlining the microridges. Full-length Ezrin was additionally found in the subapical actin layer, likely mediated by its C-terminal actin-binding domain that recognized actin filaments in the subapical compartment and/or interacted with formin family members. Concomitant with the appearance of

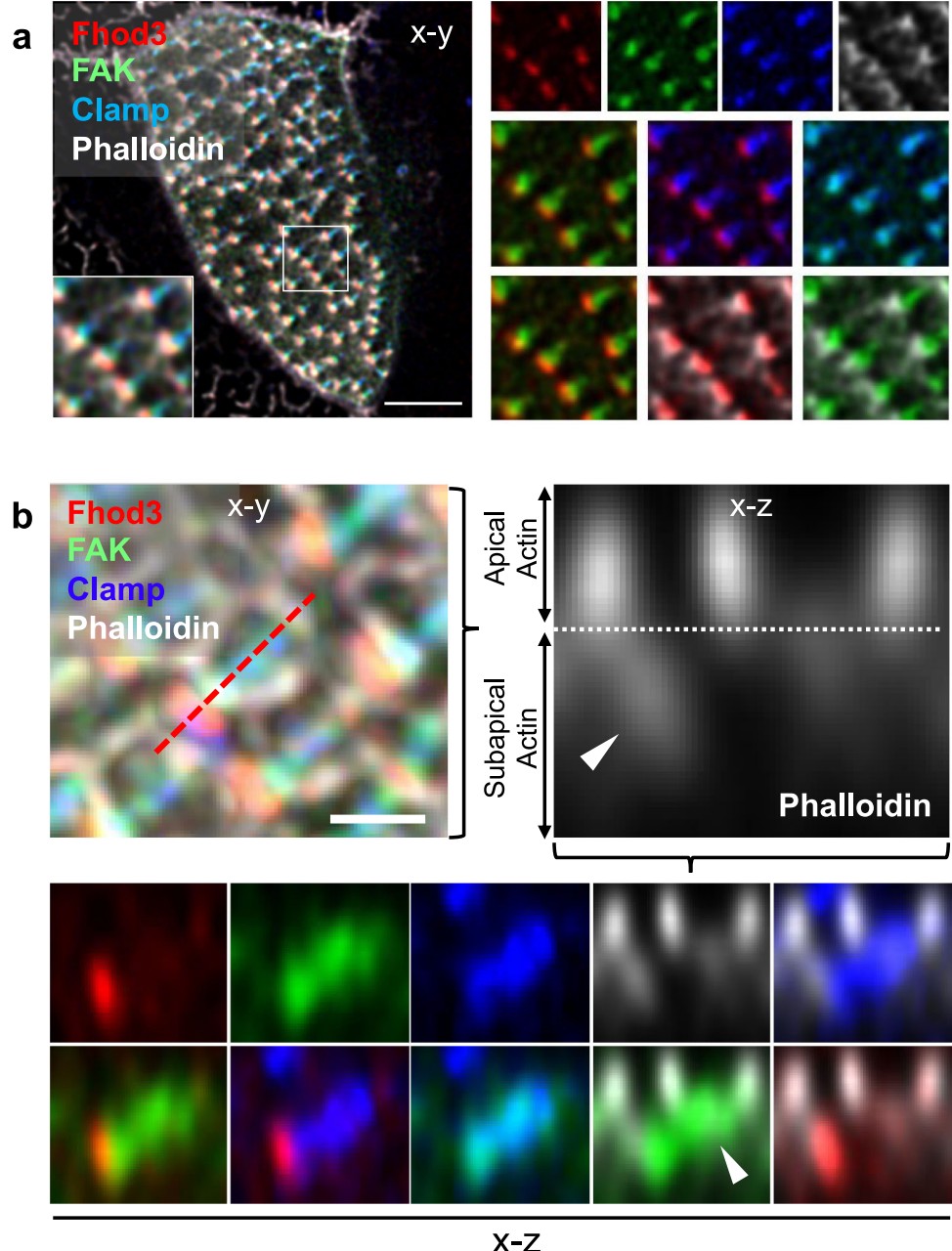

**Fig. 4 Fhod3 localizes to the tip of the anterior rootlet. a** Fhod3 (RFP, red) localizes to the distal end of the anterior rootlets (labeled with BFP-Clamp, blue) in between phalloidin (white)-labeled actin filaments and GFP-labeled FAK (scale bar, 5 µm). Magnified images depict the pair-wise comparison of fluorescent proteins. **b** The x-z projection reveals the presence of Fhod3 at the distal tip of the anterior basal body rootlet within the subapical actin layer. The left upper image depicts the maximal intensity projection of the confocal section used for 3D reconstruction (scale bar, 1 µm). The right upper picture represents a magnification of the phalloidin (white)-labeled image, depicting the border (dashed line) between apical and subapical (arrowhead) actin layer. While Fhod3 (RFP, red) labeled the tip of the anterior rootlet (BFP-Clamp, blue), FAK (GFP, green) localized to the more proximal part of the rootlet (arrowhead). Magnified images depict the pair-wise comparison of fluorescent proteins. Images were obtained at stage 29-30.

subapical actin filaments, the anterior rootlet was tilted towards the apical plasma membrane, resulting in an apparent increase in rootlet length in the x-y plane. However, in contrast to observations in *Tetrahymenia thermophila*[58], we did not observe an actual change in rootlet length. Fhod3, a formin recently implicated in apical constrictions during mouse neural tube closure[59], localized to the distal tip of the anterior rootlet, where it partially overlapped with FAK, a central component of the ciliary adhesion complex that plays a crucial role in ciliogenesis[24]. Ezrin interacts with the N-terminal domain of FAK, and contributes to its activation by supporting phosphorylation of FAK on tyrosine

397[60]. Additional studies need to elucidate how these three proteins cooperate to orchestrate the architecture of the subapical actin network. We previously noticed that depletion of Ezrin disrupts basal body docking in MCCs, while nucleation of the ciliary axoneme was largely unaffected[33]. Our findings now demonstrate that Ezrin is essential for microridge formation and rootlet anchoring; depletion of Ezrin prevents the formation of membrane protrusions, resulting in an arbitrary alignment of basal bodies relative to the apical plasma membrane. Furthermore, manipulation of Ezrin function at later stages of MCC development by light-sensitive initiation of dominant-negative

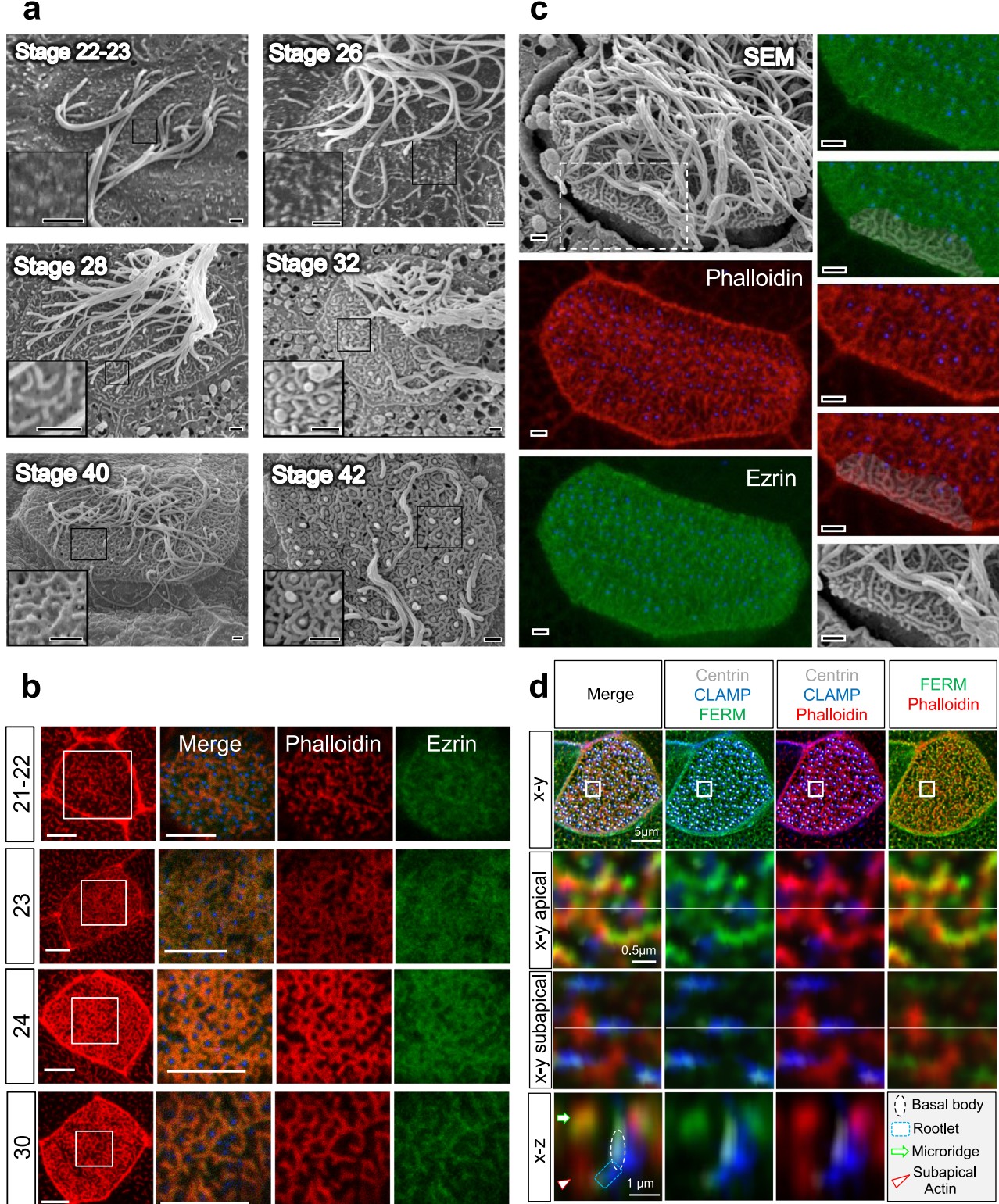

**Fig. 5 Microridge formation coincides with actin and Ezrin organization. a** Scanning electron microscopy (SEM) revealed that microridges form at stage 26, and continued to increase in complexity towards stage 40, when cilia start to retract (scale bars, 1 μm). **b** Ezrin, labeled with GFP, follows the organization of actin bundles that surround the basal bodies, labeled by BFP-Centrin in the merged image (scale bars, 5 μm). **c** Correlation between SEM and phalloidin (red)-label actin filaments revealed that the actin bundles observed in the apical actin layer follow the microridges detected by SEM (scale bars, 1 μm). **d** High-resolution confocal imaging reveals that actin bundles (phalloidin, red) arising from the anterior rootlet, connect to FERM-labeled microridges. Images were obtained at stage 29-30.

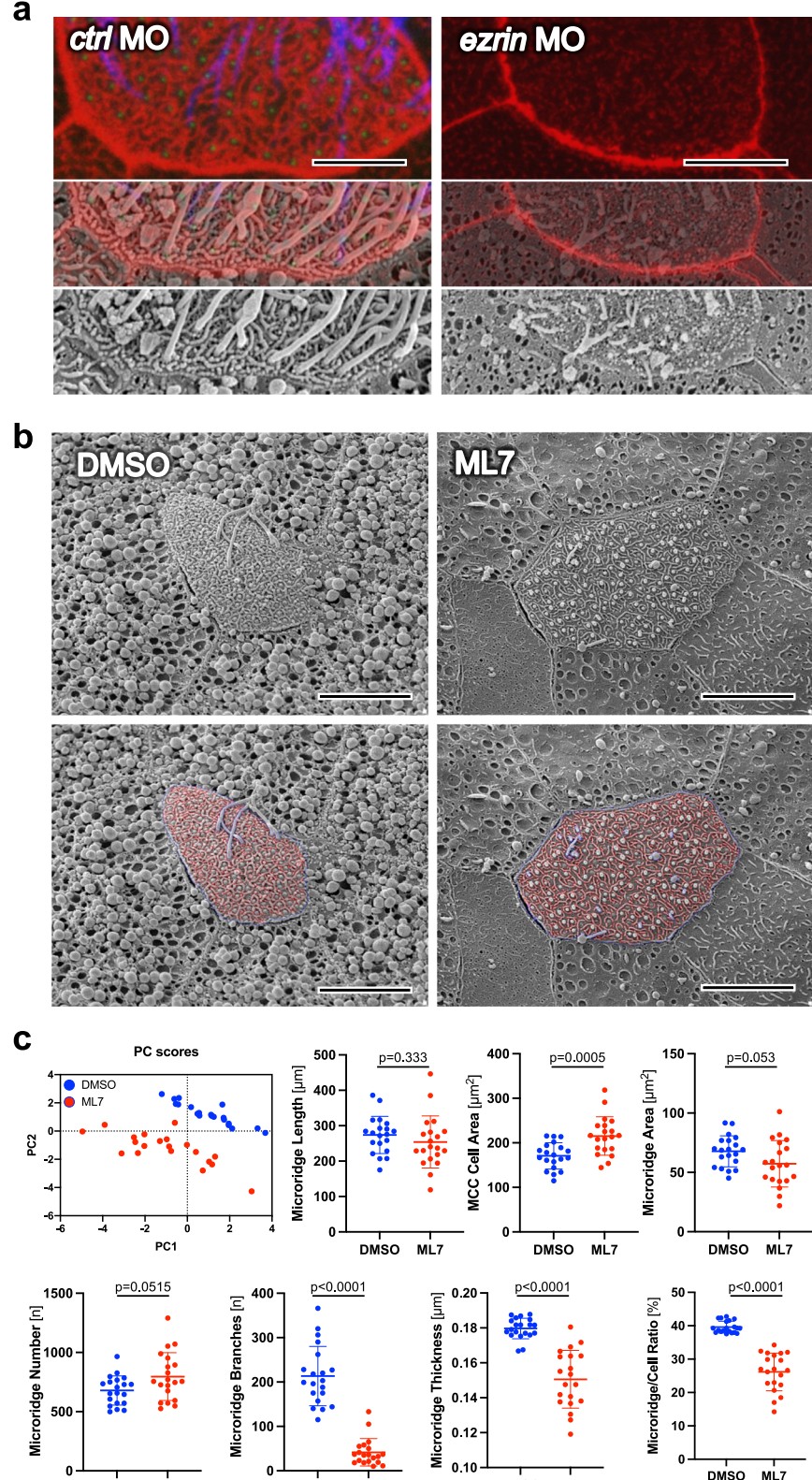

**Fig. 6 Ezrin and Nonmuscle Myosin II are essential for microridge formation. a** While the actin cytoskeleton, labeled by phalloidin (red) co-localizes with microridges visualized by scanning electron microscopy (SEM), both actin cytoskeleton and microridge architecture are completely disrupted after knockdown of *ezrin* (scale bar *ctl* MO, 4.5 μm; scale bar *ezrin* MO, 9 μm). **b** Inhibition of Myosin Light Chain Kinase (MLCK) by ML7 disrupted microridge development (scale bars, 9 μm). **c** Twenty cells obtained from 2 DMSO- or ML7 (40 μM) treated *Xenopus* embryos were visualized by SEM and analyzed by U-net[41]. Principal component analysis (PC scores) separated the DMSO- and ML7-treated MCCs based on seven different parameters (microridge length, MCC cell area, microridge area, microridge number, branches, thickness, and microridge to cell ratio). While the surface area of MCCs treated with ML7 was increased, the number of microridges, the number of branch points and the microridge thickness was significantly reduced, decreasing the ratio between cell surface area and the area occupied by microridges (mean; Mann–Whitney *U* test). Images were obtained at stage 29-30. Source data are provided as a Source Data file.

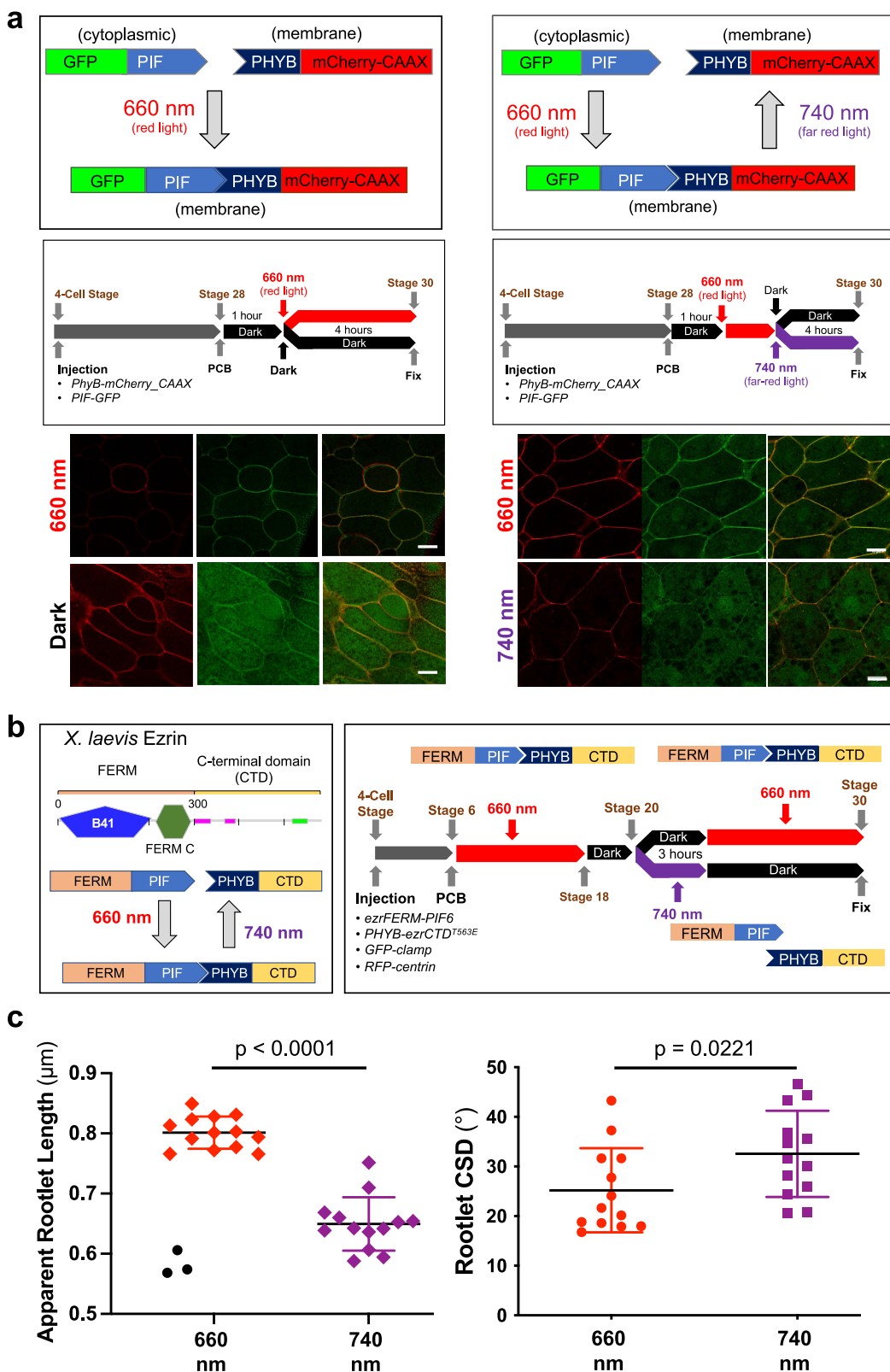

Ezrin fragments interfered with rootlet anchoring. Together, these findings demonstrate that microridges are generated during MCC development through NMII-dependent mechanisms. In mature MCCs, microridges present an essential component of the basal body docking and anchoring process. Microridges are currently a largely neglected organelle[44]. Our findings uncover a hitherto unknown function of microridges, and provide insight into the mechanisms that allow motile cilia to maintain their polarity and a coordinated fluid flow.

## Methods

**Embryo manipulations**. All experiments were approved by the local authorities (Aktenzeichen 35-9185.81/G-17/62; Regierungspräsidium Freiburg, Germany). *Xenopus laevis* embryos were obtained by in vitro fertilization[61], and staged

**Fig. 7 Optogenetic manipulation of Ezrin function in multiciliated cells of the *Xenopus* epidermis. a** To test whether the *Arabidopsis thaliana* phytochrome system mediates protein interactions in MCCs in response to red light (660 nm), the phytochrome interaction factor (PIF) fused to GFP (GFP-PIF) was co-injected with phytochrome B (PHYB) fused to mCherry, localized to the plasma membrane by the C-terminal CAAX motif of Kras (PHYB-mCherry-CAAX). In response to red light, PHYB-mCherry-CAAX recruited GFP-PIF to the plasma membrane, while GFP-PIF remained in the cytoplasm after exposure to far-red light (740 nm) (scale bars, 10 µm). **b** The N-terminal FERM domain of Ezrin (SMART, http://smart.embl-heidelberg.de/) was fused to PIF, while the C-terminal part, containing the actin-binding domain, was fused to PHYB. While red light (660 nm) maintains the Ezrin fusion, far-red right light (740 nm) creates two Ezrin fragments, generating a dominant-negative effect. mRNAs were injected at the 4-cell stage and phycocyanobilin (PCB) (15 µM) was added at stage 6. To maintain the Ezrin in an assembled state, *Xenopus* embryos were exposed to red light until stage 18. At stage 20, embryos were split into two groups. One group was exposed to far-red light for 3 h to create Ezrin fragments and then returned to dark light, maintaining the dissociation. The other group was first kept in the dark, then returned to red light, keeping the Ezrin fragments assembled. **c** Far-red light (740 nm) disrupted rootlet tilting, revealed by shortening of the apparent rootlet length, and increased the rootlet circular standard deviation (CSD), demonstrating more randomly aligned rootlets. Depicted are the results of two independent experiments, using 5 embryos per condition. The symbols display the results for individual multiciliated cells (660 nm: 16 cells; 740 nm: 13 cells). Three cells, all derived from one embryo and depicted as black circles, were identified as outliers (ROUT, Q = 1%). The significance level was $p = 0.0754$ including the three outliers, and $p < 0.0001$ excluding the outliers (Mann–Whitney $U$ test). Source data are provided as a Source Data file.

according to Nieuwkoop and Faber[62]. Microinjections were made into the two ventral blastomeres at the four- to eight-cell stage to target the epidermis. Each blastomere was injected with 10 nl of a solution containing antisense morpholino oligonucleotides (MO) (GeneTools, LLC, 1001 Summerton Way, Philomath, OR 97370, USA), and various amounts of purified mRNAs, using a time- and pressure-triggered microinjection system (Narishige International Limited, Willow Business Park, Willow Way, London SE26 4QP, U.K.). Embryos were cultured at 13–22 °C in 0.3x Marc's Modified Ringer's (MMR) solution until they reach the desired developmental stages.

**DNA reagents**. For expression in *Xenopus* embryos, *Xenopus centrin*, and *clamp*, *fhod3*, *ezrin*, FAK were subcloned into VF10 vectors with GFP, RFP, mCherry, BFP at the N- or C-terminus. Ezrin FERM domain (residues 1–310) and FHOD3 deletions N1 (residues 1–504), N2 (residues 1–733), N3 (residues 1–942), and C (residues 768–1506) were cloned by PCR. The ROCK phosphorylation sites of *ezrin* were mutated, using PCR-based site-directed mutagenesis (T563A, T563E). Plasmids were linearized and used as templates for mRNA synthesis. In vitro transcription was performed using mMessage Machine (Invitrogen); transcripts were purified using RNeasy mini (QIAGEN, Qiagene Strasse 1, 40724 Hilden). The sequences of the MOs are as follows: *control* MO (*ctrl* MO), 5'-CCTCTTACCTCAGTTACAATTTATA-3'; *fhod3* MO #1 5'-TGTGTTCTCTTT AACTTACCTTTGA-3' (splice blocking, e6i6); *fhod3* MO #2 5'- ACAAACGAAGCCATGATAAGTTCAC-3' (translation blocking); *fhod3* MO #3 5'- TTAGATCAGTTAATCCTTACCTCTT-3' (splice blocking, e5i5); *ezrin* MO 5'-CCGGTTTGGGCATTTTCACTTCTGC-3' (translation blocking). The *Xenopus laevis ezrin* FERM domain was fused to GFP and cloned into a pCS2 vector, containing the multiciliated cell-specific alpha-tubulin promoter (pCS2.*tuba1a*:MCS). Plasmids of the *Xenopus* formin family proteins were kind gifts from A. Miller (University of Michigan, USA) and T. Higashi (Fukushima Medical University, Japan). Plasmids containing *PIF6* and *PHYB-mCherryCAAX* (Plasmid #154913 created by Jonathan Clarke[63] and #51567 created by Chao Tang[64], addgene, 490 Aresenal Way, Suite 100, Watertown, MA 02472) were subcloned into VF10 vectors with *GFP*, *ezrin* FERM domain (residues 1–310), or *ezrin* C-terminal domain (CTD) (residues 297–582) containing the phospho-mimetic T563E mutation. The pCS2.*tuba1a*:ezrFERM-GFP plasmid DNA was injected at 200 pg per cell. MOs were injected at 1 pmol (*ezr* MO), 2 pmol *fhod3* MO #1, or 3 pmol *fhod3* MO #3). *Ctrl* MO concentrations were chosen accordingly. mRNA was injected using the following amounts: fluorescent protein (FP)-*centrin* 100–120 pg; FP-*clamp* 150–300 pg; FP-*fhod3* (and truncations) 200–400 pg; GFP-*FAK* 300 pg; GFP-*ezrin* 200–300 pg; *ezrin* FERM-GFP 100–200 pg; GFP-*ezrin* CTD 600 pg; GFP-*ezrin* CTD$^{T563E}$ 600 pg; GFP-$\zeta$ tubulin 600 pg; GFP-PIF6 100 pg; *PHYB-mCherryCAAX* 300 pg; *ezrFERM-PIF6* 100–200 pg; *PHYB-ezrCTD* 350–700 pg.

**Drug treatment**. ML7 hydrochloride (Tocris/Bio-Techne GmbH, Borsigstraße 7a, 65205 Wiesbaden-Nordenstadt, Germany) and SMIFH2 (Sigma-Aldrich Chemie GmbH, Eschenstraße 5, 82024 Taufkirchen, Germany) was used at a final concentration of 40 µM and 2–4 µM, respectively. After the drug was applied to the culture medium at stage 18, embryos were cultured until stage 32-35.

**Confocal microscopy**. Staining of the embryos was performed by fixing embryos at room temperature (RT) for 40 min to 1 h with MEMFA (1 M MOPS, 2 mM EGTA, 1 mM MgSO$_4$, 38% formaldehyde), followed by washing them with PBST (1x PBS, 0.1% Tween 20) three times[65]. For F-actin staining, Alexa Fluor 488 (A12379) –, Alexa Fluor 568 (A12380) –, or Alexa Fluor 647 (A22287) -Phalloidin (Invitrogen/Thermo Fisher Scientific GmbH, Im Steingrund 4-6, 63303 Dreieich, Germany) were used at a 1:500 dilution. Confocal fluorescence microscopy was performed at room temperature; 3D reconstructions were performed, using Zeiss Zen Black 2.3 (Carl Zeiss Microscopy Deutschland

GmbH, Carl-Zeiss-Strasse 22, 73446 Oberkochen, Germany), Leica LAS X 3.5.6.1594 (Leica Microsystems GmbH, Ernst-Leitz-Strasse 17–37, 35578 Wetzlar, Germany) and IMARIS (Oxford Instruments plc, Tubney Woods, Abingdon, Oxon OX13 5QX, UK) software. Images were obtained at developmental stage 29-30 unless otherwise indicated.

**Electron microscopy**. Scanning electron microscopy (SEM) was performed by fixing *Xenopus* embryos in Bouin´s solution (HT10132, Sigma-Aldrich) over night at 4 °C, followed by dehydration through a series of ascending concentrations of EtOH ethanol (EtOH) for 15 min each (70%, 80%, 90%, 100%)[66]. After incubation in 50:50 EtOH/ hexamethyldisilazan (HMDS) (CAS-# 999-97-3; Sigma-Aldrich) and 100% HMDS, the solvent was allowed to evaporate. All samples were coated with gold using a Polaron Cool Sputter Coater E 5100 (Quorum, Judges House, Lewes Road, Laughton, East Sussex, BN8 6BN, UK). Samples were imaged using a Leo 1450 VP electron microscope (Carl Zeiss). Transmission electron microscopy (TEM) was performed as described previously[33,67]. *Xenopus* embryos were fixed in 4% paraformaldehyde and 2% glutaraldehyde in 0,1 M cacodylate buffer overnight at 4 °C, washed in 0,1 M cacodylate buffer and post-fixed with 0,5% osmiumtetroxide in ddH2O for 60 min on ice. After 6 washing steps in ddH2O the tissue was en bloc stained with 1% uranyl acetate for 1 h at room temperature in the dark and dehydrated by successive incubations in 30%, 50%, 70%, 90%, 100% EtOH and 100% aceton for 15 min. After washing in 100% aceton, embryos were incubated with increasing concentrations of resin (25%, 50%, 75%) for 2 h each and left in 100% Durcupan (EMS#14020) over night without component D. The next day the samples were incubated with 100% Durcupan (all components) for 1 h and polymerized for 48 h at 60 °C. Transverse ultrathin sectioning (60 nm) was performed using a Leica EM UC7 (Leica). The sections were mounted on copper grids, and analyzed with a Zeiss LEO 906E or LEO 912 electron microscope (Carl Zeiss Microscopy Deutschland GmbH, Carl-Zeiss-Strasse 22, 73446 Oberkochen, Germany). For 3D reconstruction of single cilia, TEM pictures of 90 nm thick serial sections were aligned and reconstructed using Reconstruct Version 1.1.0.0 (https://synapseweb.clm.utexas.edu/software-0). The 3D models were tilted to measure the minimal angle between rootlet-and basal body axes, using ImageJ. The distances of the rootlet tip to the membrane were measured using Reconstruct. For correlative light-electron microscopy (CLEM), immunofluorescence microscopy and SEM were performed sequentially. Maximum intensity projections of immunofluorescent images were correlated to SEM images, using Fiji ImageJ v1.51 s software.

**Ciliary flow measurement**. For the analysis of ciliary flow, the embryos (stage 30–32) were placed in 0.3× MMR containing 0.02% of the anesthetic drug MS222 (Sigma-Aldrich) and small fluorescent beads (Invitrogen) at RT, and were analyzed by time-lapse microscopy, using a SPOT Insight FireWire system (Diagnostic Instruments) at 10x on a Leica MZ16 stereomicroscope. Acquired images were exported as TIFF files to arrange figures.

**Optogenetics**. To test the PHYB/PIF system in *Xenopus*, embryos were injected at the 4-cell stage with GFP-*PIF6* mRNA (100 pg) and *PHYB-mCherry-CAAX* mRNA (300 pg). Phycocyanobilin (PCB; Santa Cruz Biotechnology, Dallas TX, USA) (15 µM) was added at stage 28 and embryos were incubated under red light (660 nm, 10–30 µE). Subsequently, they were kept in the dark or irradiated by far-red light (740 nm, 100 µE). Embryos were fixed and the localization of GFP and mCherry were assessed by confocal microscopy. For experiments with ezrin, embryos were injected with *ezrFERM-PIF6* (100 and 200 pg), and *PHYB-ezrCT$^{T563E}$* (350 and 700 pg) mRNA as well as with plasmids encoding for *GFP-clamp*, and *RFP-centrin*. PCB (15 µM) was added at stage 6, and embryos were incubated under red light (660 nm, 10–30 µE) until stage 18. Batches were then split into two groups. The first group was incubated under far-red light (740 nm,

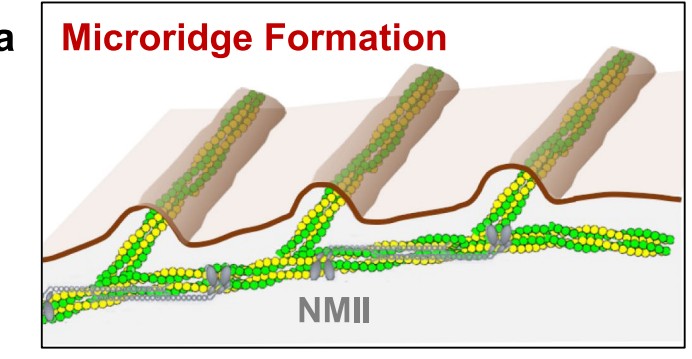

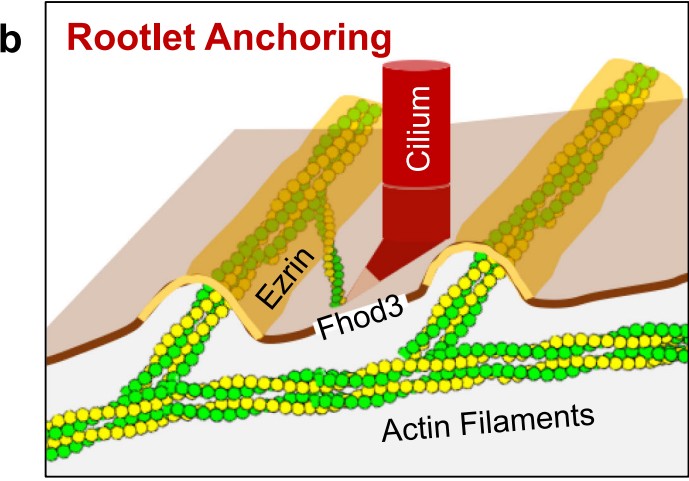

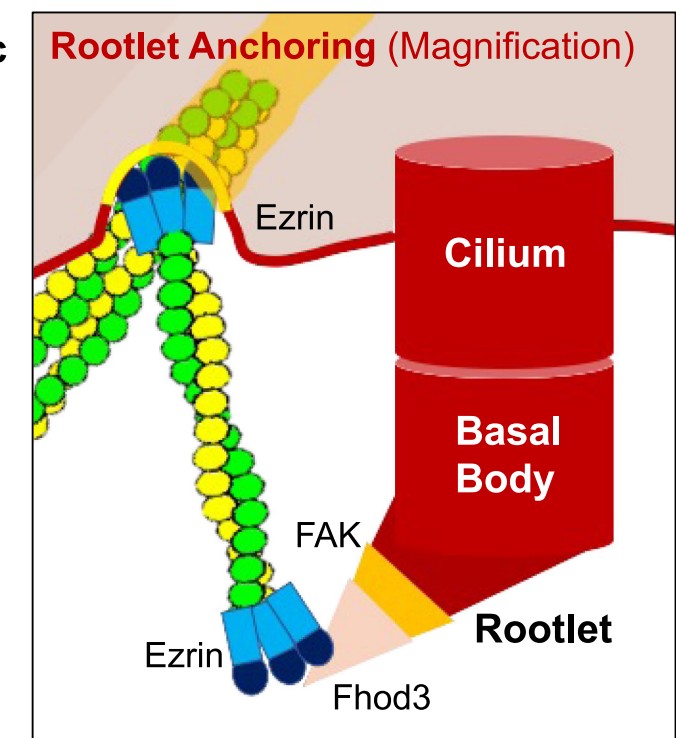

**Fig. 8 Microridge-like structures anchor basal body rootlets. a** Multiciliated cells of the *Xenopus* epidermis form microridge-like structures, requiring Nonmuscle Myosin II (NMII). **b** Actin filaments, connecting the Fhod3-labeled tip of the anterior rootlet to Ezrin-containing microridges, anchor motile cilia into their final position. **c** Fhod3 localizes adjacent to FAK to the tip of the anterior rootlet. Ezrin interacts with Fhod3 and represents part of the actin-anchoring complex at the apical plasma membrane.

100 μE) for 3 h, starting at stage 20, and kept in the dark subsequently. The second group was returned to red light after 3 h. Embryos from both groups were fixed at stage 30, and analyzed.

**Extraction of cell and microridge features from SEM images**. Cilia were removed by pH treatment to visualize microridges by SEM. Images were segmented using U-Net, a convolutional neural network[41]. As a training set, between 3 and 5 images were fully or partially annotated by manual coloration of cilia bases, remaining cilia, microridges and the area outside the MCCs. Training was performed using the ImageJ U-Net plugin, running 20.000 learning cycles at a learning rate of 1E-4. Using custom Wolfram Mathematica code, the segmented microridges were further processed by discarding small elements <0.04 μm², skeletonization and pruning of branches <90 nm. After vectorization, the microridges' lengths and number of branching points were determined. The thickness d of the microridges was estimated using the equation $A = d\,l + \left(\frac{\pi}{8} e - \frac{1}{2} k\right) d^2$, where A is the total area of all microridges, e the total number of microridge ends and k the total number of branching points.

**Cell culture, co-immunoprecipitation, and Western blots**. HEK 293T cells, routinely tested to exclude mycoplasma contamination, were transiently transfected, using the calcium phosphate method[65]. For co-immunoprecipitation, cells were lysed in 1% Triton-X, 20 mM Tris, pH7.5, 50 mM NaCl, 50 mM NaF, 15 mM $Na_4P_2O_7$, 0.1 mM EDTA, 2 mM $Na_3VO_4$ supplemented with protease inhibitor mix (Roche), incubated with 30 μl anti-Flag M2-agarose beads (A2220; Sigma-Aldrich, Eschenstrasse 5, 82024 Taufkirchen, Germany) for 2 h, and washed with lysis buffer. Precipitates were fractionated by SDS/PAGE, and analyzed by mouse monoclonal Flag antibody (1:3000) (F3165; Sigma-Aldrich) and rabbit anti-V5 (1:4000) (AB3792, Sigma-Aldrich). Uncropped Western blots are shown in Supplementary Fig. 13.

**Statistics and reproducibility**. Quantification for rootlet length were performed using ImageJ. For rootlet length and angle, the linear measurement tool of ImageJ was used to determine distance and angle between the tip of the anterior rootlet and the opposite border of the basal body. The circular standard deviation of the rootlet angles of an individual cell was calculated in Microsoft Excel using the formula

$$CSD = \sqrt{-2\ln r} \quad (1)$$

$$r = \sqrt{\left(\frac{\sum\limits_{i=1}^{n} \sin \alpha}{n}\right)^2 + \left(\frac{\sum\limits_{i=1}^{n} \cos \alpha}{n}\right)^2}. \quad (2)$$

Ciliary flow was quantified using IMARIS (Oxford Instruments plc, Tubney Woods, Abingdon, Oxon OX13 5QX, UK)). All confocal images are representative of at least 30 different cells from 10 different embryos. Microsoft Excel, GraphPad Prism and RStudio were used for statistical tests and graphs, depicting mean and standard deviation unless otherwise indicated. For statistical significance, the nonparametric Mann–Whitney $U$ test was used. The data analysis for rootlet length was performed using generalized linear model (GLM) or generalized linear mixed models (GLMM). All experiments were repeated independently at least two times using at least three individual embryos each time. Depicted symbols represent the mean of approximately 150 measurements per multiciliated cell unless otherwise indicated.

Accession numbers. *Xenopus ezrin* NM_001093923.1, *Gallus FAK* NM_205435.1, UniGene accession numbers: *Xenopus centrin4* NM_001096664.1, *Xenopus clamp*, NM_001096937.1.

**Reporting summary**. Further information on research design is available in the Nature Research Reporting Summary linked to this article.

## Data availability
The authors declare that all data supporting the findings of this study are available within the article and its supplementary information files, or from the corresponding author upon reasonable request. All data generated in this study are provided in the Source Data file. Source data are provided with this paper.

## Code availability
Binary versions of the U-Net caffe extensions for Ubuntu Linux 16.04 are available at https://lmb.informatik.uni-freiburg.de/resources/opensource/unet/[41].

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

## Acknowledgements

We would like to thank Roland Nitschke and the Life Imaging Center at the University of Freiburg, and Séverine Kayser for excellent technical support. We gratefully acknowledge Professor Ann L. Miller, University of Michigan, and Professor Tomohito Higashi, Fukushima Medical University, for providing the GFP-tagged formin constructs. This study was supported by the Deutsche Forschungsgemeinschaft (DFG, German Research Foundation): WA 597/20-1 (to G.W.), Project-ID 431984000 – SFB 1453 (to P.W and G.W.), Germany's Excellence Strategy (CIBSS_EXC-Project ID 390939984 to P.W. and G.W.), WA 3365/2-1 (Emmy-Noether Program to P.W.), and by the Else-Kröner Fresenius Stiftung (to G.W.).

## Author contributions

T.Y. and G.W. conceived and analyzed the experiments. T.Y. J.W., and M.B. performed and analyzed the *Xenopus* experiments. A.P. and C.E. performed immunoprecipitation assays, and provided technical assistance. M.H. and D.E. performed and analyzed the TEM and SEM experiments. Ö.C., O.R., and T.B. analyzed actin dynamics, M. U. performed the U-net analysis, G.R. helped to implement the optogenetics experiments, P.W. provided conceptual input and revised the manuscript. T.Y. and G.W. wrote the manuscript with input from all authors.

## Funding

## Competing interests

The authors declare no competing interests.
