## [Peer Review File · Nature Communications]

Microridge-like structures anchor motile ciliaEditorial Note: Parts of this Peer Review File have been redacted as indicated to remove third-party material where no permission to publish could be obtained.

Reviewers' Comments:

Reviewer #1:

Remarks to the Author:

The manuscript from Yasunaga et al. addresses the formation of an elaborate actin network in multiciliated cells. They show interesting localization of actin nucleating proteins at the tip of rootlets and show actin projecting towards the apical surface and helping to connect rootlets to the apical actin network. They show evidence that the apical actin forms prior to the sub-apical actin, which is important as the sub-apical pool is known to be involved in generating cilia polarity. They present beautiful data showing distinct and intriguing localization of various formins and propose that these are important for linking the rootlets to FAK. They show that the apical actin becomes increasingly organized between ST21-30 and that this coincides with an increase in ezrin. They describe Ca based fluctuations in actin and cell size via contractions. They go on to show that ezrin and NMII are both critical for the formation of actin in MCCs.

There is a lot of beautiful data in this paper and I think with some tweaks it can be a very nice paper. However, I have several fairly serious issues with some of their claims. Perhaps my biggest issue is with the claim that they are even studying microridges. While microridges are poorly studied, they have been described in more generic epithelial cells with significantly less actin. The authors state microridges have been observed in other ciliated epithelia such as the fallopian tubes. However, in this work it is NOT the MCCs that have microridges. MCCs have tremendous amount of actin that is quite distinct from other epithelial cells and while there may be some resemblance to microridges, I suspect that there are also important differences. I think it makes more sense to describe this actin network for what it is and then mention that it shares features of microridges. They claim that the subapical actin can't form without the microridges but they are using treatments that will likely disrupt all actin structures so the causality is questionable and in fact the "microridges" aren't really prevalent until Stage 28 after the sub-apical actin appears at least partially formed. While the calcium results are really cool, the results are not well integrated with the story. Is there any relationship between the contractions and the proper formation of the actin network. Finally, there is a consistent lack of experimental detail that makes much of the data challenging to interpret and understand.

Specific comments:

Figure 2 c. The figure in 2c makes it hard to tell if the posterior rootlet is orthogonal, which seems important if one wants to make the claim that it is labeling this. While the extended data might be better, it is not described in enough detail to know what one is looking at. It doesn't quite look like side projections.

Figure 2 d. I think a better description is needed. The FAK staining does not match the literature, where FAK is reported to be at both ends of the rootlet. But here it seems as if the apical part of the cell is removed from the image, as we only see the subapical actin. Is this correct? If so it should be clearly explained that the apical part of the cell has been removed from the image. This would explain why the FAK does not look right. A zoomed in side projection of the presented data could also be really helpful.

The tilting of the rootlets is a very interesting observation that is surely important. The authors clearly appreciate (and state) that the tilting of the rootlet makes it appear longer in x-y projections. Yet later in the paper they show the effect of various factors on "rootlet length". Unless this was done with very careful and challenging 3D measurements I think that the descriptions of length change are misleading and the terminology should be dramatically changed. I have seen no evidence in this paper that the length of rootlets has changed. In all the experiments with rootlet length it is unclear how the measurements were made and what exactly are the authors claim. If the analysis was done in two dimensions on stacked images then the length decrease could represent a shortening of the rootlet or could represent a failure to tilt and extend laterally which seems more likely. There is some interesting work (which should probably be cited) on ciliary rootlet length from tetrahymena (e.g. Galati et al

2014), which seem different from this, so it is important to clarify.

It is unclear to me in the experiments in Figure 4 how the cells were identified as MCCs. Particularly in b and c there is a fairly similar amount of red staining in analyzed cells and their neighbors. While it is not labeled what the red is I am guessing actin (utrophin) which would suggest that these experiments were not done on MCCs. This should be made explicitly clear, and if they are claiming that the cells are MCCs, how are they identified.

As the authors note, there are dramatic changes to the actin network and dynamics at different stages. However in most of the experiments it is not stated what ST the analysis was done which makes it really hard to interpret or assimilate the information.

Minor.... 2c the c is misplaced. 2d the images are not aligned well.

Figure 4. All minor issues, but ...the ST is described as 28 but labeled as 29. The representative images should be remade without the imbedded text or it should be cropped out and the time should be labeled as min in the bottom left corner. Images should be their own subpanel with better description in the legend. Finally the description for the bottom graphs describes only the red Ca++ line not the blue.

Reviewer #2:

Remarks to the Author:

The manuscript by Yasunaga et al entitled "Microridge-like structures anchor motile cilia" unravels the mechanism of cilia anchorage. It reports how subapical actin filaments connect the ciliary rootlets to microridges, which is required for their orientation. Based on their data, the authors conclude that the function of Ezrin is essential for the formation of microridges and rootlet anchorage, which is essential for ciliogenesis. This is an elegant study that reports the novel function of microridges in cilia anchorage, makes an attempt to uncover the anchorage mechanism, and as such significantly contributes towards our understanding of the process of ciliogenesis. However, while the description of the process and causality are well established by performing the high-quality imaging analyses, the functional analysis involving Ezrin doesn't conclusively prove that the link between rootlets and microridges via formin-Ezrin is essential for the orientation of the ciliary rootlets. This is possibly the most exciting aspect of this work and needs to be substantiated further.

Major comment:

The functional analysis clearly indicates that the loss of microridges due to the depletion of Ezrin has a profound effect on cilia formation. However, this could be a secondary consequence (or a non-specific effect). Authors need to show that severing the link between microridges and rootlets, while keeping the microridges intact, disrupts ciliogenesis. This would prove the point that anchorage is indeed essential. Can authors make a construct of Ezrin that does not interact with Fhod3, express that in the Ezrin deficient cells, and show that while microridge formation gets rescued, the rootlet orientation phenotype persists and ciliogenesis remains perturbed? Alternatively, a loss of function analysis for Fhod3 using morpholino (with all specificity controls) may help to prove the point if the Fhod3 morphants do not show the absence of microridges but selectively exhibit ciliogenesis defects. Some of these experiments will conclusively prove the point that the anchorage via Fhod3 (Subapical Actin)-Ezrin is indeed functionally relevant.

Minor comments:

1. Both the Ezrin knockdown and ML7 treatments are performed at the organismal level and there is a possibility of cell non-autonomous effect leading to reported phenotypes (in microridge organization, cilia orientation). It would be important to test if Ezrin knockdown or Myosin inhibition alter microridges and ciliogenesis in a cell-autonomous manner in multi-ciliated cells; for example, by performing clonal injections of Ezrin morpholino or using photoactivable blebbistatin.
2. In the experiment presented in Extended figure 10, the Ezrin deficiency should be complemented with the expression of full-length wild-type control Ezrin. This will be a good control and also prove the specificity of the morpholino phenotype.
3. The top schematic in Extended Data Fig 12 is highly speculative at this stage and should be removed. In the last schematic, it's not clear where Ezrin and FhoD3 localize. It also doesn't reflect the fact that Ezrin and FhoD3 interact with each other.

Reviewer #3:

Remarks to the Author:

The paper by Yasunaga et al. on "Microridge-like structures anchor motile cilia" attempts to show how cilia are oriented in the multiciliated cells of the *Xenopus* epidermis. Overall, the paper presents nice observations on Calcium burst or microridge organizations. However, the different observations are not really linked together and thus do not bring an understanding of the mechanisms. Data are often over-interpreted and quantifications are sparse and sometime inappropriate. For all these reasons, I do not think it is suitable as it stands for publication in Nature Communications.

Major points:

1) In the first part, the authors described the actin structures that appear sequentially during the formation of the motile cilia. Others have already made these observations (Ioannou, A., et al. 2013; Werner, M. E., et al. 2011; Mahuzier, A., et al. 2018) but the author did not mention them. The definition of apical and subapical actin structures is quite difficult to apprehend. In other studies, the apical structure was defined as the structure underneath the apical membrane (encompassing the distal part of the centriole, stained by centrin) while the subapical actin was defined as the actin structure at $\sim 0.5\mu\text{m}$ underneath at the level of the proximal part of the centrioles (so Centrin negative). Sticking with this definition would help readers.

The image resolution notably in Z is low and in my opinion co-localization data over-interpreted. Using immunofluorescence data, the authors suggest that at mature stages, the rootlet is tilted and anchored to the apical actin structure (Figure 1). However, this is not what is observed on the EM data on the extended Figure 4. It is thus confusing and should be addressed using higher resolution microscopy or EM.

2) The authors show a co-localization of subapical actin with the rootlet and Fhod3 (Figure 1 and 2 and extended Figure 2). However, co-localization data are not sufficient to assess their interdependencies. Removing the rootlets or Fhod3 should be attempted before to conclude whether nucleation of the subapical actin is dependent on them. Especially, as treatment with a drug inhibiting formin activity (SMIFH2) seems to remove the apical and not subapical actin (Extended Figure 1). The role of Fhod3 in the orientation of the cilia could also be assessed.

3) Centriole orientation is usually addressed by analyzing the position of the rootlet according to the centriole and the plan of the tissue, not by measuring the length of the rootlet. The authors if they really want to understand the mechanisms of centriole orientation should thus perform this kind of quantifications. Alternatively, the decrease of the length might be an indication of rootlet structural defects that might be worth investigating.

4) The authors carry on with the localization of Ezrin. The link with the previous observations are quite

weak. Once again, co-localization observations are not sufficient to assess whether ezrin is indeed important to anchor the ciliary rootlets (Figure 3). The fact that ciliary rootlets are shorter in Ezrin morphants does not mean there is an anchorage issue. Actually, Ezrin Mo perturbs centriole migration toward the plasma membrane by affecting the actin organization (Epting, D., et al. 2015). The migration defect might then affect the maturation of the centrioles and the formation of the rootlets. Rootlets are important for centriole orientation (Kim, S. K., et al. 2018). Thus, the role of the microridge is not clear in this paper.

5) Microridge formation was shown to depend on cortical contraction and Myosin. The authors are able to reproduce this result and to perturb to some extent the formation of the microridge by using drugs against the myosin light chain kinase (Figure 5). However, it is not clear whether only the microridge are affected or the whole actin network (Extended Figure 11). There is no attempt to quantify how actin is affected. The centriolar phenotypes analyzed are only the length of the rootlets (no mention of the number of cells quantified) without any attempt to look at migration, maturation or even orientation of the centriole. Therefore, it is not clear whether microridge perturbation affect centriole orientation.

6) Finally, the authors observed bursts of Calcium in some cells on the epithelium (Figure 4, extended figure 9 and videos), that is associated with either constriction or expansion of the cells. It is perfectly unclear to me how the authors are sure to analyze multiciliated cells. The utrophine staining of the cells that constrict is not reminiscent of the actin staining in multiciliated cells. There is several cell types on the epithelium including ionocytes and small secretory cells that also intercalate (Drysdales and Elinson, 1992; Quigley, et al., 2011; Dubaissi et al., 2014). On the cells that undergo expansion the utrophine staining might suggest multiciliated cells, but then there is no constriction. In any case, the authors did not check whether the induced constriction promote microridge formation in this system nor if it change the orientation of the centrioles.

Minor points:

- The authors looked at the 15 formins it would then be of interest to tell whether the other formins are located in interesting patterns.
- As the actin structures change during the process of multiciliation, it is important to mention the age at which the different immuno staining have been performed.

REVIEWER COMMENTS

We would like to thank all three referees for their highly professional criticisms. We are grateful for their comments that helped to significantly improve the manuscript.

Response to Reviewer #1:

The manuscript from Yasunaga et al. addresses the formation of an elaborate actin network in multiciliated cells. They show interesting localization of actin nucleating proteins at the tip of rootlets and show actin projecting towards the apical surface and helping to connect rootlets to the apical actin network. They show evidence that the apical actin forms prior to the sub-apical actin, which is important as the sub-apical pool is known to be involved in generating cilia polarity. They present beautiful data showing distinct and intriguing localization of various formins and propose that these are important for linking the rootlets to FAK. They show that the apical actin becomes increasingly organized between ST21-30 and that this coincides with an increase in ezrin. They describe Ca based fluctuations in actin and cell size via contractions. They go on to show that ezrin and NMII are both critical for the formation of actin in MCCs.

There is a lot of beautiful data in this paper and I think with some tweaks it can be a very nice paper. However, I have several fairly serious issues with some of their claims. Perhaps my biggest issue is with the claim that they are even studying microridges. While microridges are poorly studied, they have been described in more generic epithelial cells with significantly less actin. The authors state microridges have been observed in other ciliated epithelia such as the fallopian tubes. However, in this work it is NOT the MCCs that have microridges. MCCs have tremendous amount of actin that is quite distinct from other epithelial cells and while there may be some resemblance to microridges, I suspect that there are also important differences. I think it makes more sense to describes this actin network for what it is and then mention that it shares features of microridges. They claim that the subapical actin can't form without the microridges but they are using treatments that will likely disrupt all actin structures so the causality is questionable and in fact the "microridges" aren't really prevalent until Stage 28 after the sub-apical actin appears at least partially formed. While the calcium results are really cool, the results are not well integrated with the story. Is there any relationship between the contractions and the proper formation of the actin network. Finally, there is a consistent lack of experimental detail that makes much of the data challenging to interpret and understand.

We agree with the reviewer that the membrane protrusions on the surface of multiciliated cells of the *Xenopus* epidermis appear quite different from the microridges formed on the zebrafish epidermis. We have therefore titled the manuscript "Microridge-like structures anchor motile cilia", and now use this term throughout the Abstract and Introduction. We underline that the term "microridge" is used for simplicity in the reminder of the manuscript (page 3), and added a paragraph in the Discussion to highlight potential differences between the microridge-like structures of MCCs and the microridges described of the surface of the zebrafish epidermis.

However, we would like to point out that microridges/microplacae can assume a "myriad" of different structures ¹. We also noted that microridges are quite sensitive to the relative harsh fixation methods used to prepare samples for SEM. The peculiar structure of zebrafish microridges may endow them with a relative resistance to standard fixation methods, making them to a unique model to study microridges. However, gentler fixation methods in combination with high resolution scanning (e.g. helium ion scanning) may uncover microridges/microplacae of different architectures on many more cell surfaces ².

Specific comments:

Figure 2 c. The figure in 2c makes it hard to tell if the posterior rootlet is orthogonal, which seems important if one wants to make the claim that it is labeling this. While the extended data might be

better, it is not described in enough detail to know what one is looking at. It doesn't quite look like side projections.

Depicted are diagonal views after 3D reconstruction of confocal images, using the software IMARIS. The figure legend was extended accordingly (now Fig. 3d).

Figure 2 d. I think a better description is needed. The FAK staining does not match the literature, where FAK is reported to be at both ends of the rootlet. But here it seems as if the apical part of the cell is removed from the image, as we only see the subapical actin. Is this correct? If so it should be clearly explain that the apical part of the cell has been removed from the image. This would explain why the FAK does not look right. A zoomed in side projection of the presented data could also be really helpful.

Figure 2d (now Fig. 4) was extended, providing 3D reconstructions of confocal images that depict the localization of Fhod3 in relationship to FAK within the subapical actin layer.

The tilting of the rootlets is a very interesting observation that is surely important. The authors clearly appreciate (and state) that the tilting of the rootlet makes it appear longer in x-y projections. Yet later in the paper they show the effect of various factors on "rootlet length". Unless this was done with very careful and challenging 3D measurements I think that the descriptions of length change are misleading and the terminology should be dramatically changed. I have seen no evidence in this paper that the length of rootlets has changed. In all the experiments with rootlet length it is unclear how the measurements were made and what exactly are the authors claim. If the analysis was done in two dimensions on stacked images then the length decrease could represent a shortening of the rootlet or could represent a failure to tilt and extend laterally which seems more likely. There is some interesting work (which should probably be cited) on ciliary rootlet length from tetrahymenia (e.g. Galati et al 2014), which seem different from this, so it is important to clarify.

We agree with the reviewer that this is a central observation, and have therefore decided to perform systematic 3D EM reconstructions, using transmission electron microscopy in combination with serial sections. A total of twenty rootlets were analyzed, ten at stage 20 and ten at stage 32. After 3D reconstruction, the distance between the tip of the rootlet and the plasma membrane was measured. This approach revealed a significant decrease of the distance between the plasma membrane and the tip of the rootlet, confirming rootlet tilting during the development of the multi-ciliated cells of the *Xenopus* epidermis (new Figure 2).

Galati et al. ³ describes that lengthening of kinetodesmal fibers (KF) plays an important role in response to temperature changes and mechanical stress in *T. thermophila*, highlighting the importance of rootlet-like structures in maintaining ciliary functions. The analysis was performed, using x-y confocal images. Since the KF are not aligned parallel to the cell surface (Galati et al., Fig. 1G), this analysis cannot entirely exclude that the observed KF lengthening is also partially caused by changes of the angle between basal bodies and KFs.

Nevertheless, we now emphasize in the **Discussion** that the tilting (with an apparent increase in rootlet length monitored in the x-y plane) does not, in contrast to observations in *T. thermophila*, refer to an actual change in rootlet length, and cite the paper by Galati et al. as proposed by the reviewer.

It is unclear to me in the experiments in Figure 4 how the cells were identified as MCCs. Particularly in b and c there is a fairly similar amount of red staining in analyzed cells and their neighbors. While it is not labeled what the red is I am guessing actin (utrophin) which would suggests that these

experiments were not done on MCCs. This should be made explicitly clear, and if they are claiming that the cells are MCCs, how are they identified.

While it might be difficult to recognize MCCs in Fig. 4 (now Fig. 6) due to the high imaging speed and low image resolution, the dense apical actin of MCCs was clearly detectable when recording the images. The enclosed image, taken with higher resolution, demonstrates that MCCs can be readily identified, using fluorescently marker RFP-utrophin. Please see also Fig. 4 in Kulkarni et al., comparing α -tubulin staining with fluorescent utrophin⁴. We now mention RFP-utrophin in the figure legend, explaining the red-labeled structures in Fig. 6.

[Redacted]

Fig.: Identification of multi-ciliated cells by RFP-utrophin.

The left images demonstrates that multi-ciliated cells can readily be identified by their prominent RFP-Utrophin staining. In the right panel (Fig. 4B in⁴) Kulkarni et al. showed that the cells with prominent Utrophin staining are also positive for acetylated α -tubulin.

As the authors note, there are dramatic changes to the actin network and dynamics at different stages. However in most of the experiments it is not stated what ST the analysis was done which makes it really hard to interpret or assimilate the information.

Images were generally obtained at developmental stage 29-30 unless otherwise specified. We have added to **Methods** that images were obtained at developmental stage 29-30 unless otherwise specified. We have labeled figures accordingly.

Minor.... 2c the c is misplaced. 2d the images are not aligned well.

We have corrected the misplacement and re-aligned the images in 2d

Figure 4. All minor issues, but ...the ST is described as 28 but labeled as 29. The representative images should be remade without the imbedded text or it should be cropped out and the time should be labeled as min in the bottom left corner. Images should be their own subpanel with better description in the legend. Finally the description for the bottom graphs describes only the red Ca⁺⁺ line not the blue.

We have made all changes as requested: the imbedded text was removed, the time is now labeled in minutes, and subpanels were added.

Response to Reviewer #2:

The manuscript by Yasunaga et al entitled “Microridge-like structures anchor motile cilia” unravels the mechanism of cilia anchorage. It reports how subapical actin filaments connect the ciliary rootlets to microridges, which is required for their orientation. Based on their data, the authors conclude that the function of Ezrin is essential for the formation of microridges and rootlet anchorage, which is essential for ciliogenesis. This is an elegant study that reports the novel function of microridges in cilia anchorage, makes an attempt to uncover the anchorage mechanism, and as such significantly contributes towards our understanding of the process of ciliogenesis. However, while the description of the process and causality are well established by performing the high-quality imaging analyses, the functional analysis involving Ezrin doesn't conclusively prove that the link between rootlets and microridges via formin-Ezrin is essential for the orientation of the ciliary rootlets. This is possibly the most exciting aspect of this work and needs to be substantiated further.

Major comment:

The functional analysis clearly indicates that the loss of microridges due to the depletion of Ezrin has a profound effect on cilia formation. However, this could be a secondary consequence (or a non-specific effect). Authors need to show that severing the link between microridges and rootlets, while keeping the microridges intact, disrupts ciliogenesis. This would prove the point that anchorage is indeed essential. Can authors make a construct of Ezrin that does not interact with Fhod3, express that in the Ezrin deficient cells, and show that while microridge formation gets rescued, the rootlet orientation phenotype persists and ciliogenesis remains perturbed? Alternatively, a loss of function analysis for Fhod3 using morpholino (with all specificity controls) may help to prove the point if the Fhod3 morphants do not show the absence of microridges but selectively exhibit ciliogenesis defects. Some of these experiments will conclusively prove the point that the anchorage via Fhod3 (Subapical Actin)-Ezrin is indeed functionally relevant.

Thank you very much for these thoughtful suggestions. We agree that severing the link between microridges and rootlets without disrupting microridges would conclusively demonstrate the significance of the proposed anchorage mechanism.

Following the reviewer's suggestions, we performed *fhod3* knockdown experiments, using three different *fhod3* morpholino oligonucleotides (MOs). Depletion of *fhod3* caused defects in rootlet tilting and polarization (new Supplemental Fig. 4).

Since the MO-based *fhod3* depletion may affect early developmental stages, we decided to additionally employ an opto-genetical approach to disrupt the anchorage between basal body rootlets and microridges at a later developmental stage. Over-expression of Ezrin truncations exert dominant-negative effects (Supplementary Fig. 11 and 12). Utilizing this observation, we created an Ezrin fusion protein that dissociates upon exposure to far red light (740 nm), but re-associates at red light (660 nm), using the phycocyanobilin-dependent PHYB/PIF system⁵. To establish this approach in the *Xenopus* epidermis, GFP was fused to PIF, and PHYB to a membrane-associated mCherry version (mCherry-CAAX). Upon exposure to red light, the GFP-PIF fusion protein was correctly recruited to the plasma membrane, while exposure to far red light reversed the interaction of both proteins (new Fig. 8a). Fusing the N-terminal FERM domain of Ezrin to PIF, and the C-terminal domain (CTD) to PHYB, we then used this system to first dissociate Ezrin at stage 20 (exposure to far red light), and subsequently to either maintain the dissociation by leaving the *Xenopus* embryos in the dark, or by re-exposing the multi-ciliated cells to red light, which triggers the assembly of both fragments, abolishing the dominant negative effect (new Fig. 8b). This approach worked very nicely, and demonstrated that even at later developmental stages, a disruption of the ezrin-

based anchorage affects rootlet length and polarity. To our knowledge, this is the first time that optogenetics has been applied to study multi-ciliated cells of the *Xenopus* epidermis.

Minor comments:

1. Both the Ezrin knockdown and ML7 treatments are performed at the organismal level and there is a possibility of cell non-autonomous effect leading to reported phenotypes (in microridge organization, cilia orientation). It would be important to test if Ezrin knockdown or Myosin inhibition alter microridges and ciliogenesis in a cell-autonomous manner in multi-ciliated cells; for example, by performing clonal injections of Ezrin morpholino or using photoactivable blebbistatin.

The referee is raising an interesting point. We added experimental data, expressing an N-terminal truncation of ezrin (*ezrFERM-GFP*) under the control of the α -tubulin promoter specifically in MCCs. Microinjection of *ezrFERM-GFP* at the 4-cell stage resulted in a highly mosaic expression pattern at stage 30 (new Supplemental Fig. 12), allowing us to analyze neighboring MCCs with or without expression of GFP-labeled N-terminal Ezrin. Moderate expression altered rootlet length and polarity, while stronger expression of the FERM domain resulted in ciliogenesis defects with reduction of apical actin in a cell-autonomous manner.

2. In the experiment presented in Extended figure 10, the Ezrin deficiency should be complemented with the expression of full-length wild-type control Ezrin. This will be a good control and also prove the specificity of the morpholino phenotype.

We agree that mRNA rescue experiments are standard controls for MO phenotypes. However, the rescue of PCP as well as many cytoskeletal phenotypes has been notoriously difficult since both, excess or lack of any of the components result in dominant negative effects, disrupting developmental processes.

Rescuing the ezrin MO phenotype with wild-type *ezrin* proved to be similarly challenging. We provide the data from a successful experiment for the referee (see attached Figure), but prefer to not include the rescue experiment in the manuscript because co-expression of *ezrin* mRNA also occasionally increased the severity of the phenotype.

3. The top schematic in Extended Data Fig 12 is highly speculative at this stage and should be removed. In the last schematic, it's not clear where Ezrin and FhoD3 localize. It also doesn't reflect the fact that Ezrin and FhoD3 interact with each other.

Following the reviewer's suggestion, we have removed the speculative aspects from the top schematic (formerly Extended Data Fig. 12, now Fig. 9a), and added an insert, displaying the interaction between Ezrin and Fhod3.

Response to Reviewer #3:

The paper by Yasunaga et al. on “Microridge-like structures anchor motile cilia” attempts to show how cilia are oriented in the multiciliated cells of the *Xenopus* epidermis. Overall, the paper presents nice observations on Calcium burst or microridge organizations. However, the different observations are not really linked together and thus do not bring an understanding of the mechanisms. Data are often over-interpreted and quantifications are sparse and sometime inappropriate. For all these reasons, I do not think it is suitable as it stands for publication in Nature Communications.

Major points:

1) In the first part, the authors described the actin structures that appear sequentially during the formation of the motile cilia. Others have already made these observations (Ioannou, A., et al. 2013; Werner, M. E., et al. 2011; Mahuzier, A., et al. 2018) but the author did not mention them. The definition of apical and subapical actin structures is quite difficult to apprehend. In other studies, the apical structure was defined as the structure underneath the apical membrane (encompassing the distal part of the centriole, stained by centrin) while the subapical actin was defined as the actin structure at ~0.5µm underneath at the level of the proximal part of the centrioles (so Centrin negative). Sticking with this definition would help readers.

We apologize for omitting the publications of Ioannou, A. et al, 2013, and Mahuzier A. et al. 2018. We have added them to the manuscript; the paper by Werner, M.E. et al., 2011 was already cited in the previous version.

MCCs are not always perfectly plane, creating overlapping centrin-positive apical and subapical regions. However, the structures of the apical and subapical actin web are distinct, allowing their separation. We also added an image in Fig. 4b to clarify the distinction between the apical and subapical actin layer.

The image resolution notably in Z is low and in my opinion co-localization data over-interpreted. Using immunofluorescence data, the authors suggest that at mature stages, the rootlet is tilted and anchored to the apical actin structure (Figure 1). However, this is not what is observed on the EM data on the extended Figure 4. It is thus confusing and should be addressed using higher resolution microscopy or EM.

[Redacted]

Figure: Schematic representation of microridges based on EM and immunolocalization analyses (from Pinto C.S. et al., 2019).

TEM analysis does not permit the identification of single actin filaments or bundles projecting from rootlets to the plasma membrane. Using high-resolution EM tomography, Pinto C.S. et al., 2019 demonstrated that the actin cytoskeleton of microridges represents an extremely complex network of fibers⁶.

The important information depicted in our TEM analysis is the regular spacing of membrane protrusions next to each ciliary axoneme. We extended our TEM analysis, and performed 3D reconstructions of basal body rootlets to confirm the tilting of the basal body rootlet during the development of multiciliated cells (new Figure 2).

2) The authors show a co-localization of subapical actin with the rootlet and Fhod3 (Figure 1 and 2 and extended Figure 2). However, co-localization data are not sufficient to assess their interdependencies. Removing the rootlets or Fhod3 should be attempted before to conclude whether nucleation of the subapical actin is dependent on them. Especially, as treatment with a drug inhibiting formin activity (SMIFH2) seems to remove the apical and not subapical actin (Extended Figure 1). The role of Fhod3 in the orientation of the cilia could also be assessed.

As requested, we depleted Fhod3, using three different morpholino oligonucleotides. Fhod3 deficiency was associated with changes in rootlet lengthening and rootlet polarization (rootlet circular standard deviation, CSD). The new data is now presented in Supplementary Fig. 4.

3) Centriole orientation is usually addressed by analyzing the position of the rootlet according to the centriole and the plan of the tissue, not by measuring the length of the rootlet. The authors if they really want to understand the mechanisms of centriole orientation should thus performed this kind of quantifications. Alternatively, the decrease of the length might be an indication of rootlet structural defects that might be worse investigating.

We agree with the Reviewer that the analysis of rotational polarity requires the combined analysis of centriole and rootlet position. However, our aim was to demonstrate that rootlet anchorage is associated with tilting and apparent lengthening of rootlets. Since we wanted to assess the individual cell's coherency in aligning basal body rootlets, we used changes of rootlet length to detect abnormal anchorage.

4) The authors carry on with the localization of Ezrin. The link with the previous observations are quite weak. Once again, co-localization observations are not sufficient to assess whether ezrin is indeed important to anchor the ciliary rootlets (Figure 3). The fact that ciliary rootlets are shorter in Ezrin morphants does not mean there is an anchorage issue. Actually, Ezrin Mo perturbs centriole migration toward the plasma membrane by affecting the actin organization (Epting, D., et al. 2015). The migration defect might then affect the maturation of the centrioles and the formation of the rootlets. Rootlets are important for centriole orientation (Kim, S. K., et al. 2018). Thus, the role of the microridge is not clear in this paper.

We agree that morpholino oligonucleotide-based depletion of essential ciliary components can affect early unrelated developmental processes. As the reviewer pointed out, defective basal body migration could influence ciliogenesis and actin organization. To avoid early developmental perturbations, we used optogenetics to disrupt the anchorage between basal body rootlets and microridges after stage 20. As shown in -Fig. 8, we created an Ezrin fusion protein that dissociates upon exposure to far red light (740 nm), but re-associates at red light (660 nm) (Figure 8). This approach revealed that even at later developmental stages, the disruption of ezrin function affects rootlet length and polarity.

5) Microridge formation was shown to depend on cortical contraction and Myosin. The authors are able to reproduce this result and to perturb to some extent the formation of the microridge by using drugs against the myosin light chain kinase (Figure 5). However, it is not clear whether only the microridge are affected or the whole actin network (Extended Figure 11). There is no attempt to quantify how actin is affected. The centriolar phenotypes analyzed are only the length of the rootlets (no mention of the number of cells quantified) without any attempt to look at migration, maturation or even orientation of the centriole. Therefore, it is not clear whether microridge perturbation affect centriole orientation.

As demonstrated in Figures 5 and Supplementary Figures 6 and 7, the cortical actin network co-localizes with Ezrin-positive microridges. The changes of the microridges, and therefore changes of the cortical actin network, were extensively quantified in Fig. 7.

The role of Ezrin in basal body migration and ciliogenesis has been extensively studied (see for example Epting D. et al., 2015 ⁷). Studies that disrupt early events of ciliogenesis may indirectly affect basal body anchorage, and thus may be difficult to interpret. Furthermore, the optogenetic approach also demonstrated that defects arise after apical basal body migration.

6) Finally, the authors observed bursts of Calcium in some cells on the epithelium (Figure 4, extended figure 9 and videos), that is associated with either constriction or expansion of the cells. It is perfectly unclear to me how the authors are sure to analyze multiciliated cells. The utrophine staining of the cells that constrict is not reminiscent of the actin staining in multiciliated cells. There is several cell types on the epithelium including ionocytes and small secretory cells that also intercalate (Drysdale and Elinson, 1992; Quigley, et al., 2011; Dubaissi et al., 2014). On the cells that undergo expansion the utrophine staining might suggest multiciliated cells, but then there is no constriction. In any case, the authors did not check whether the induced constriction promote microridge formation in this system nor if it change the orientation of the centrioles.

We agree that it can be difficult to recognize MCCs in Fig. 4 due to the high imaging speed and the resulting low resolution. However, the dense apical actin of MCCs was clearly detectable while recording these images. The enclosed image demonstrates that MCCs can be readily identified, using fluorescently marked utrophin. Please see also Fig. 4 in Kulkarni et al., comparing α -tubulin staining with fluorescent utrophin ⁴. We now mention RFP-utrophin in the figure legend to explain the red-labeled structures (now Figure 6).

[Redacted]

Fig.: Identification of multi-ciliated cells by RFP-utrophin.

The left images demonstrates that multi-ciliated cells can readily be identified by their prominent RFP-Utrophin staining. In the right panel (Fig. 4B in ⁴) Kulkarni et al. showed that the cells with prominent Utrophin staining are also positive for acetylated α -tubulin.

Minor points:

- The authors looked at the 15 formins it would then be of interest to tell whether the other formins are located in interesting patterns.

We have previous published the localization of Daam proteins (Yasunaga T. et al., 2015) ⁸. Fhod1, Fhod3, and Delphinin (included in the manuscript) displayed very specific localizations within the ciliary compartment. Most of the other Formins only co-localized weakly with ciliary structures or the apical actin network (see attached data).

- As the actin structures change during the process of multiciliation, it is important to mention the age at which the different immuno staining have been performed.

We apologize for the omission. We have now included the stages at which the analyses were performed.

References

1. Depasquale JA. Actin Microridges. *Anat Rec (Hoboken)* **301**, 2037-2050 (2018).
2. Rice WL, *et al.* High resolution helium ion scanning microscopy of the rat kidney. *PLoS One* **8**, e57051 (2013).
3. Galati DF, *et al.* DisAp-dependent striated fiber elongation is required to organize ciliary arrays. *J Cell Biol* **207**, 705-715 (2014).
4. Kulkarni SS, Griffin JN, Date PP, Liem KF, Jr., Khokha MK. WDR5 Stabilizes Actin Architecture to Promote Multiciliated Cell Formation. *Dev Cell* **46**, 595-610 e593 (2018).
5. Kramer MM, Lataster L, Weber W, Radziwill G. Optogenetic Approaches for the Spatiotemporal Control of Signal Transduction Pathways. *Int J Mol Sci* **22**, (2021).
6. Pinto CS, Khandekar A, Bhavna R, Kiesel P, Pigo G, Sonawane M. Microridges are apical epithelial projections formed of F-actin networks that organize the glycan layer. *Sci Rep* **9**, 12191 (2019).
7. Epting D, *et al.* The Rac1 regulator ELMO controls basal body migration and docking in multiciliated cells through interaction with Ezrin. *Development* **142**, 1553 (2015).
8. Yasunaga T, *et al.* The polarity protein Inturned links NPHP4 to Daam1 to control the subapical actin network in multiciliated cells. *J Cell Biol* **211**, 963-973 (2015).

Addendum

	ctrl MO	ezrin MO	ezrin rescue
Embryos analyzed	4	4	4
Cells analyzed	10	14	12
Rootlets analyzed	1821	2238	1947
Average Rootlet Length [μm]	0.792	0.695	0.776

Figure: Co-injection of *ezrin* mRNA rescues the shortening of rootlet length, caused by the knockdown of *ezrin*. Control (*ctrl*) or *ezrin* morpholino oligonucleotides (MO) (1 pmol/0.1 mM) were co-injected with 150 pg GFP-*Clamp*/100 pg RFP-*Centrin* mRNA to determine rootlet length. To rescue the *ezrin*-deficiency, 200 pg *ezrin* mRNA was added. Multi-ciliated cells were imaged at stage 29-30. The number of analyzed embryos, cells, and rootlets are depicted in the table. Each symbol in the graph represents the average of one cell.

Figure: Localization of Formins within multi-ciliated cells. Only Fhod1, Fhod3 (see manuscript) and Delphinin displayed a specific expression pattern in multi-ciliated cells.

Reviewers' Comments:

Reviewer #1:

Remarks to the Author:

The revision from Yasunaga et al represents a significant improvement over the original submission. The addition of the optogenetic Ezrin experiments are quite nice and add significantly to the rigor. I was and remain supportive of this paper. However, I was a little disappointed in the lack of responsiveness to some of the original critiques. I appreciate the authors response to my critique about rootlet length and they have appropriately added a sentence stating that the tilting is measured with "apparent" rootlet length. They have also addressed this in the discussion in some detail. However, all of the graphs are still labeled as "rootlet length" when that is really not what is being measured. While I appreciate that the authors are trying to keep the graphs simple, it is simply not accurate. I would urge them to label the axis as lateral distance or even apparent rootlet length. Anything that accurately reflects what is being measured. Second, multiple reviewers commented on the lack of integration of the calcium imaging with the rest of the story. That has not really changed. The technique is super cool and I have no doubt that Ca spikes are causing contraction. The issue is what do contractions have to do with microridges and rootlets which is not really addressed. I personally think the rest of the story stands on its own and that this data only confuses things. I would urge them to remove that part, and save it for a later paper where they can actually show the significance. As it stands it only detracts from an otherwise beautiful paper.

Reviewer #2:

Remarks to the Author:

The revised version of the manuscript by Yasunaga et al has addressed most of the concerns that I had raised earlier. Importantly, they have included functional analyses involving fhod3 depletion and disrupting Ezrin via an elegant optogenetic approach. Besides, they have also performed cell-autonomous perturbation Ezrin function. These additional experiments strengthen the importance of Ezrin and FhoD3 and the link between them in cilia anchoring and orientation. The following points can be clarified (to the Editor) before the final acceptance.

1. How rootlet CSD is quantified has not been included in the methods section
2. For comparisons of CSD and rootlet length, t-test statistics has been used. The Authors should confirm (and state in the methods) that the normality test was performed before conducting the t-test. This is important since the distribution seems to be skewed in some of the experiments, e.g. 8C (CSD), S4B (CSD for MO1 in rescue expt)

Reviewer #3:

Remarks to the Author:

Some major results have been added to the manuscript from Yasunaga et al, especially the optogenetic analysis of Ezrin disruption at late stage of development. However, most of the major comments I made previously, have not been properly addressed.

The calcium experiments are still not properly integrated in the paper; While it is interesting to know that there is some Calcium burst during MCC maturation, it is not clear whether it promotes actin reorganization, myosin activation, microridge formation, rootlet apparent lengthening or anything else unrelated to the story. Even the contraction reported is only observed in 50% of the cells with a burst, so is it a direct consequence? This part should be removed from the paper as well as the modelling that is speculative.

The authors still speak about centriole orientation or lengthening throughout the paper, while they only look at an apparent lengthening that is a consequence of rootlet titling, at least in the wild type condition. Terminology is thus misleading and must be changed.

The authors did Fhod3 morpholino to answer whether Fhod3 is involved in nucleating subapical actin. However, they did not look at all at actin organization following the morpholino. As drugs against formins perturbs both the apical and subapical actin, we thus wonder whether Fhod3 is specific of the subapical actin.

Concerning the myosin experiment, it is still not clear whether only microridges are disturbed by ML7 treatment and whether this disruption are a cause of the increase of SD of centriole tilting.

Finally, the number of cells analyzed in the last experiment on ezrin optogenetic constructs is quite low and I am not sure of the pertinence of using a t-test in this condition.

Minor comments

Figure 1 and 2 could be merged as well as Figure 3 and 4.

Figure S2 there is no indication of the orientation of the images and it is actually extremely difficult to know what we are looking at.

RESPONSE TO THE REVIEWER COMMENTS

Reviewer #1 (Remarks to the Author):

The revision from Yasunaga et al represents a significant improvement over the original submission. The addition of the optogenetic Ezrin experiments are quite nice and add significantly to the rigor. I was and remain supportive of this paper. However, I was a little disappointed in the lack of responsiveness to some of the original critiques. I appreciate the authors response to my critique about rootlet length and they have appropriately added a sentence stating that the tilting is measured with “apparent” rootlet length. They have also addressed this in the discussion in some detail. However, all of the graphs are still labeled as “rootlet length” when that is really not what is being measured. While I appreciate that the authors are trying to keep the graphs simple, it is simply not accurate. I would urge them to label the axis as lateral distance or even apparent rootlet length. Anything that accurately reflects what is being measured.

Following the reviewer’s suggestion, the axis label was changed to “Apparent Rootlet Length” in all pertinent figures.

Second, multiple reviewers commented on the lack of integration of the calcium imaging with the rest of the story. That has not really changed. The technique is super cool and I have no doubt that Ca spikes are causing contraction. The issue is what do contractions have to do with microridges and rootlets which is not really addressed. I personally think the rest of the story stands on its own and that this data only confuses things. I would urge them to remove that part, and save it for a later paper where they can actually show the significance. As it stands it only detracts from an otherwise beautiful paper.

Following the reviewers’ suggestion, we removed the calcium data from the manuscript (Fig. 6, Suppl. Fig. 9, and Suppl. Fig. 10).

Reviewer #2 (Remarks to the Author):

The revised version of the manuscript by Yasunaga et al has addressed most of the concerns that I had raised earlier. Importantly, they have included functional analyses involving rhod3 depletion and disrupting Ezrin via an elegant optogenetic approach. Besides, they have also performed cell-autonomous perturbation Ezrin function. These additional experiments strengthen the importance of Ezrin and FhoD3 and the link between them in cilia anchoring and orientation. The following points can be clarified (to the Editor) before the final acceptance.

1. How rootlet CSD is quantified has not been included in the methods section

We have added a description in the Methods section, and are also including the raw data for all figures.

2. For comparisons of CSD and rootlet length, t-test statistics has been used. The Authors should confirm (and state in the methods) that the normality test was performed before conducting the t-test. This is important since the distribution seems to be skewed

in some of the experiments, e.g. 8C (CSD), S4B (CSD for MO1 in rescue expt)

For consistency, we changed all statistics to the nonparametric Mann-Whitney-U test). All p-values were recalculated accordingly.

Reviewer #3 (Remarks to the Author):

Some major results have been added to the manuscript from Yasunaga et al, especially the optogenetic analysis of Ezrin disruption at late stage of development. However, most of the major comments I made previously, have not been properly addressed.

The calcium experiments are still not properly integrated in the paper; While it is interesting to know that there is some Calcium burst during MCC maturation, it is not clear whether it promotes actin reorganization, myosin activation, microridge formation, rootlet apparent lengthening or anything else unrelated to the story. Even the contraction reported is only observed in 50% of the cells with a burst, so is it a direct consequence? This part should be removed from the paper as well as the modelling that is speculative.

Following the reviewers' suggestion, we removed the calcium data from the manuscript (Fig. 6, Suppl. Fig. 9, and Suppl. Fig. 10).

The authors still speak about centriole orientation or lengthening throughout the paper, while they only look at an apparent lengthening that is a consequence of rootlet titling, at least in the wild type condition. Terminology is thus misleading and must be changed.

Following the reviewers' suggestion, the axis label was changed to "Apparent Rootlet Length" in all pertinent figures.

The authors did Fhod3 morpholino to answer whether Fhod3 is involved in nucleating subapical actin. However, they did not look at all at actin organization following the morpholino. As drugs against formins perturbs both the apical and subapical actin, we thus wonder whether Fhod3 is specific of the subapical actin.

Fig. 3 shows that Fhod3 localizes to the subapical actin, but some Fhod3 is also present within the apical actin layer. We have added a new Suppl. Fig. 5, demonstrating that *fhod3* depletion primarily affects the subapical actin layer. Since the organization between basal bodies, apical and sub-apical actin has to be strictly coordinated between each other for normal morphology and function, it is impossible to manipulate the function of one component without also affecting the others.

Concerning the myosin experiment, it is still not clear whether only microridges are disturbed by ML7 treatment and whether this disruption are a cause of the increase of SD of centriole tilting.

ML7 is a very potent, selective myosin light chain kinase inhibitor with a K_i value of 300 nM (K_i values for PKA and PKD are 21 and 42 μ M, respectively). Our experiments demonstrate that ML7 affects the formation of microridges (Fig. 6) and basal body polarization (Suppl. Fig. 12). The apical actin cytoskeleton and microridges form before nucleation of the subapical actin

(Fig. 1a), while basal polarization of the basal bodies correlates with the appearance subapical actin (Fig. 1f). Although MLCK inhibition has been implicated in other cellular functions (e.g., tight junction formation), the role of MLCK in microridge development, using MLCK inhibitors, has been extensively characterized^{1, 2, 3, 4}. Therefore, it appears plausible that the disruption of the microridges causes defective basal body polarization. However, as the reviewer points out, we cannot infer causality, and have therefore added a comment in the Discussion.

Finally, the number of cells analyzed in the last experiment on ezrin optogenetic constructs is quite low and I am not sure of the pertinence of using a t-test in this condition.

For consistency, we changed all statistics to the nonparametric Mann-Whitney-U test. All p-values were recalculated accordingly.

Minor comments

Figure 1 and 2 could be merged as well as Figure 3 and 4.

We agree, but after removing the calcium data, there is no need to merge these figures. However, the figures can be merged upon editorial request.

Figure S2 there is no indication of the orientation of the images and it is actually extremely difficult to know what we are looking at.

We apologize, and have added an explanation.

References

1. van Loon AP, Erofeev IS, Goryachev AB, Sagasti A. Stochastic contraction of myosin minifilaments drives evolution of microridge protrusion patterns in epithelial cells. *Mol Biol Cell* **32**, 1501-1513 (2021).
2. van Loon AP, Erofeev IS, Maryshev IV, Goryachev AB, Sagasti A. Cortical contraction drives the 3D patterning of epithelial cell surfaces. *J Cell Biol* **219**, (2020).
3. Raman R, Damle I, Rote R, Banerjee S, Dingare C, Sonawane M. aPKC regulates apical localization of Lgl to restrict elongation of microridges in developing zebrafish epidermis. *Nature communications* **7**, 11643 (2016).
4. Lam PY, Mangos S, Green JM, Reiser J, Huttenlocher A. In vivo imaging and characterization of actin microridges. *PLoS One* **10**, e0115639 (2015).

Reviewers' Comments:

Reviewer #3:

Remarks to the Author:

The manuscript on « Microridge-like structures anchor motile cilia » by Yasunaga et al. have been greatly improved since the first submission.

Only minor details remain to be addressed:

- 1) There is still discussion and methods on experiments that have been removed from the manuscript, please correct.
- 2) Having the age of the animals on each figure as well as when drugs were applied will facilitate the reading.
- 3) There is some typo mistake such as rootled instead of rootlets.

Response to Reviewer #3:

REVIEWERS' COMMENTS

Reviewer #3 (Remarks to the Author):

The manuscript on « Microridge-like structures anchor motile cilia » by Yasunaga et al. have been greatly improved since the first submission.

Only minor details remain to be addressed:

- 1) There is still discussion and methods on experiments that have been removed from the manuscript, please correct.
- 2) Having the age of the animals on each figure as well as when drugs were applied will facilitate the reading.
- 3) There is some typo mistake such as rootled instead of rootlets.

We have removed all parts of the discussion and methods related to the calcium experiments that were eliminated from the manuscript.

We have included the age of the embryos in each figure legend, and the time of drug application.

We have corrected the typos.